# ADAGC: IMPROVING TRAINING STABILITY FOR LARGE LANGUAGE MODEL PRETRAINING

## ABSTRACT

Loss spikes remain a persistent obstacle in large-scale language model pretraining. Empirically, such spikes can be triggered by a mixture of factors, including data outliers, hardware or transient computational faults, numerical precision issues, and hyperparameter settings. Regardless of the underlying cause, these spikes manifest as unstable optimizer updates, as abnormal gradients contaminate both first- and second-moment states. In this paper, we do not attempt to identify the precise root causes. Instead, we adopt a gradient-centric remedy and propose AdaGC, an adaptive, per-tensor gradient clipping scheme that prevents such contamination by bounding gradient norms relative to a tensor-wise EMA of their historical (clipped) values. AdaGC is optimizer-agnostic, requires negligible memory, and reduces communication costs compared to GlobalGC, particularly under hybrid parallel distributed training. We prove that Adam with AdaGC preserves the standard non-convex convergence rate. On Llama-2 7B, Mixtral 8×1B, and ERNIE 10B-A1.4B models, AdaGC robustly eliminates training instabilities, reducing the spike score to zero for all models, and improves downstream accuracy compared to GlobalGC by +1.32%, +1.27%, and +2.48%, respectively. Furthermore, AdaGC composes well with Muon and Lion optimizers, consistently yielding higher average accuracy and zero spike scores.

## 1 INTRODUCTION

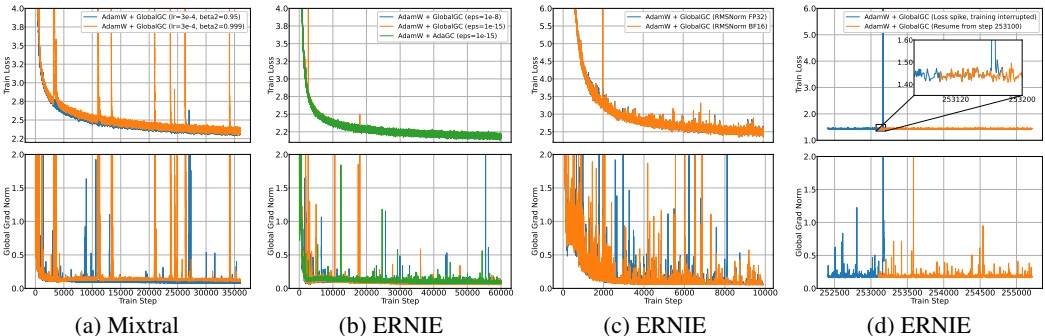

| (a) Mixtral | (b) ERNIE | (c) ERNIE | (d) ERNIE |

Figure 1: **Reproduced cases of loss spikes and mitigation via resuming.** Loss spikes are triggered by (a) increasing $\beta_2$ or (b) reducing $\epsilon$ in AdamW, (c) using lower-precision RMSNorm, even under global gradient clipping, and (d) are resolved by resuming due to stochasticity in FlashAttention backward passes.

The rapid scaling of large language models (LLM) has introduced new challenges in pretraining stability, often manifesting as abrupt loss spikes or transient divergences across a wide range of model architectures and data scales (Chowdhery et al., 2023; Touvron et al., 2023; Liu et al., 2024; Team et al., 2025; Baidu-ERNIE-Team, 2025). Despite extensive empirical studies, the fundamental causes of these instabilities remain elusive. Recent research, alongside our own analyses, indicates that loss spikes can arise from a variety of sources, including: (i) data quality issues (Chowdhery et al., 2023); (ii) hardware or transient computational faults (Su, 2025); (iii) variations in numerical precision (for example, FP32 typically offers greater robustness than BF16, whereas FP8 can sometimes enhance stability by suppressing outlier values via implicit quantization (Han, 2024; Liu et al., 2024)); and (iv) the selection of optimizer and layer normalization hyperparameters, such as

the $\epsilon$ parameter in RMSNorm or AdamW, and $\beta_2$ in AdamW (Ma et al., 2021; Cattaneo & Shigida, 2025; Bai et al., 2025). For instance, we observe that increasing $\beta_2$ or decreasing $\epsilon$ in AdamW can trigger loss spikes, whereas increasing the precision of RMSNorm from BF16 to FP32 significantly improves stability. Figure 1 presents several representative cases we have reproduced.

Although the upstream causes of instability are diverse and often subtle, these events consistently converge at the optimizer level, manifesting as abnormal gradients. Such outlier gradients are incorporated into the optimizer's first- and second-moment estimates, thereby corrupting parameter updates and propagating instability through subsequent training. Notably, we find that even resuming interrupted training (while keeping the random seed and data unchanged) can mitigate a loss spike, merely due to the stochastic nature of $dQ$, $dK$, and $dV$ in FlashAttention (Dao, 2023) (see Figure 1d). This observation further suggests that, in certain model states, even minute numerical differences can trigger a loss spike, with gradient outliers playing a critical role in both the initiation and propagation of instabilities during optimizer state updates.

While the slight stochasticity introduced by FlashAttention can sometimes circumvent a loss spike, repeatedly interrupting and resuming training imposes substantial computational overhead. Given that these instabilities stem from diverse upstream causes but ultimately converge at the optimizer level, our work *does not attempt to identify the precise root causes*. Instead, we adopt a **gradient-centric** perspective: irrespective of the initial trigger, loss spikes consistently arise when outlier gradients contaminate the optimizer states. Therefore, by preventing such gradients from entering the first- and second-moment accumulators, we provide a unified and effective strategy to mitigate training instability.

A standard mitigation strategy is global gradient clipping (GlobalGC), which bounds the global $\ell_2$ norm of the aggregated gradient. However, this approach is fundamentally mismatched to modern large-scale pretraining in two key respects: (1) *Temporal mismatch:* The optimal global clipping threshold typically decreases over the course of training; a fixed threshold risks under-clipping in later phases. (2) *Spatial mismatch:* Gradient statistics and rare spikes vary asynchronously across different parameter tensors, making a single global threshold insufficient—protecting one tensor may under-serve or over-constrain others.

To address these challenges, we introduce *Adaptive Gradient Clipping based on Local Gradient Norm* (AdaGC): a simple, per-tensor clipping rule that leverages an EMA of each tensor's historical gradient norm as a reference. Each tensor's gradient is clipped relative to its own EMA, preventing transient outliers from contaminating the first- and second-moment accumulators and, ultimately, the parameter updates. A brief warm-up period applies global clipping and initializes the EMA to avoid early overestimation. AdaGC is optimizer-agnostic and can be seamlessly integrated with AdamW, Lion, and Muon. Our main contributions are as follows:

- **A unified, gradient-centric perspective**: We clarify how loss spikes universally propagate via abnormal gradients polluting optimizer states, irrespective of their origin, motivating intervention at the gradient level prior to moving-average accumulation.
- **An adaptive, per-tensor clipping rule**: By tracking each tensor's gradient norm statistics with an EMA, AdaGC provides both temporal adaptivity and spatial specificity, suppressing outliers while minimally disturbing typical learning dynamics.
- **System efficiency and theoretical guarantees**: We analyze computational and communication overhead, showing that AdaGC reduces communication relative to GlobalGC under hybrid parallel distributed training, and we prove that Adam+AdaGC maintains an $O(1/\sqrt{T})$ convergence rate under standard non-convex conditions.
- **Empirical validation at scale**: On Llama-2 7B, Mixtral $8\times1$B, and ERNIE 10B-A1.4B models, AdaGC robustly eliminates training instabilities and improves accuracy compared to GlobalGC by +1.32%, +1.27%, and +2.48%, respectively. The method is similarly effective with AdamW, Lion, and Muon optimizers.

## 2  RELATED WORK

**Stability in large-scale pretraining:** Dozens of approaches address instability during large-model pretraining, including: architectural advances (Pre-LN Xiong et al. (2020), RMSNorm (Zhang & Sennrich, 2019)), careful initialization (Nguyen & Salazar, 2019; Takase et al., 2023; Nishida et al.,

Table 1: Comparison of major gradient/update clipping methods for training stability in pretraining. Here, $\theta_t$ denotes the model parameters, $g_t$ the gradients, $\Delta_t$ the optimizer update, $v_t$ the second momentum, $\eta_t$ the learning rate, $\lambda_{abs}$ the absolute threshould, and $\lambda_{rel}$ the relative threshold.

| Method | Algorithm | Gradient | Update | Granularity | Threshold Type |
|---|---|---|---|---|---|
| GlobalGC (Pascanu et al., 2013) | $\min\{1.0, \lambda_{abs}\frac{1}{\|g_t\|}\}$ | ✓ | ✗ | Global | Fixed constant |
| ClipByValue | $clamp(-\lambda_{abs}, \lambda_{abs})$ | ✓ | ✗ | Element | Fixed constant |
| AGC (Brock et al., 2021) | $\min\{1.0, \lambda_{rel}\frac{\|\theta_t\|}{\|g_t\|}\}$ | ✗ | ✓ | Unit | Weight $\ell_2$ norm |
| Clippy (Tang et al., 2023) | $\min\{1.0, \min(\frac{\lambda_{rel}\|\theta_t\|_\infty + \lambda_{abs}}{\eta_t * \|\Delta_t\|_\infty})\}$ | ✗ | ✓ | Tensor | Weight $\ell_\infty$ norm |
| SPAM (Huang et al., 2025) | $\text{sign}(g_t) \cdot \sqrt{\lambda_{rel} v_t}$ | ✓ | ✗ | Element | Local (vector) variance |
| LAMB (You et al., 2019) | $\frac{\phi(\|\theta_t\|)}{\|\Delta_t\|}$ | ✗ | ✓ | Tensor | Weight $\ell_2$ norm |
| **AdaGC (ours)** | $\min\{1.0, \lambda_{rel}\frac{\gamma_{t-1,i}}{\|g_{t,i}\|}\}$ $\gamma_{t,i} = \beta\gamma_{t-1,i} + (1-\beta)\|g_{t,i}\|$ | ✓ | ✗ | Tensor | EMA of gradient norm |

2024), auxiliary loss terms (Max-z loss (Yang et al., 2023)). Recent work OLMo et al. (2024) also explores combining multiple stabilization strategies. These measures improve average stability but do not directly prevent abnormal gradients from corrupting optimizer states.

**Gradient/Update Clipping:** Gradient and update clipping achieve stability by limiting the magnitude of gradients and parameter updates, preventing excessively large weight updates. Global gradient clipping (Pascanu et al., 2013) is prevalent, with innovative approaches like AGC (Brock et al., 2021) and Clippy (Tang et al., 2023), which use model weights to adjust the clipping threshold. The SPAM (Huang et al., 2025) method stabilizes the model training process by introducing a momentum reset mechanism and an element-wise gradient clipping strategy based on second-moment estimation. Alternatives like Adafactor (Shazeer & Stern, 2018), StableAdamW (Wortsman et al., 2023), and LAMB (You et al., 2019) offer update clipping techniques better suited for stability training of large-scale models. Nonetheless, a significant number of loss spikes still occur during the training of large language models, even with the application of these methodologies. Due to our gradient-centric perspective, we focus our discussion on *clipping-based* methods. These methods fall into two categories: *value-based* approaches, which truncate individual gradient components exceeding a predefined limit, and *norm-based* approaches, which rescale the entire gradient vector only when its overall magnitude exceeds a threshold. AdaGC belongs to the norm-based category, leveraging adaptive per-tensor norm thresholds to stabilize training. For a comparative summary, see Table 1.

## 3 MOTIVATION: FROM ROOT-CAUSE DIVERSITY TO A UNIFIED GRADIENT-CENTRIC REMEDY

Through a series of experiments (see Figure 1 and Figure 2), we observe that loss spikes encountered under diverse settings consistently coincide with abrupt fluctuations in the gradient norm. Comparative analyses further reveal limitations of existing methods such as GlobalGC, AGC, and Clippy: GlobalGC's static global threshold cannot detect or suppress localized abnormal gradients, allowing outliers to contaminate optimizer states and trigger instability. AGC and Clippy focus on controlling parameter updates, leaving internal moments vulnerable to large gradient outliers.

As discussed in the Introduction (Section 1), loss spikes typically result from a combination of multiple factors. While the specific triggers may vary, these loss spikes share a common manifestation: abnormally large gradients are incorporated into the optimizer's moment estimates, leading to unstable updates. Based on these analyses, we propose a unified remedy: ***regardless of the root cause, instability in large-scale training is best addressed via gradient-centric clipping.*** Specifically, only localized and adaptive clipping, applied *before* gradients are integrated into the optimizer's moment estimates, can effectively constrain the influence of outlier gradients. We thus distill two key principles for loss spike mitigation: *(1) Locality:* clip gradients for each parameter tensor individually, avoiding the insensitivity of a global threshold; *(2) Adaptivity:* dynamically adjust each tensor's clipping threshold, e.g., using an EMA of its recent gradient norms.

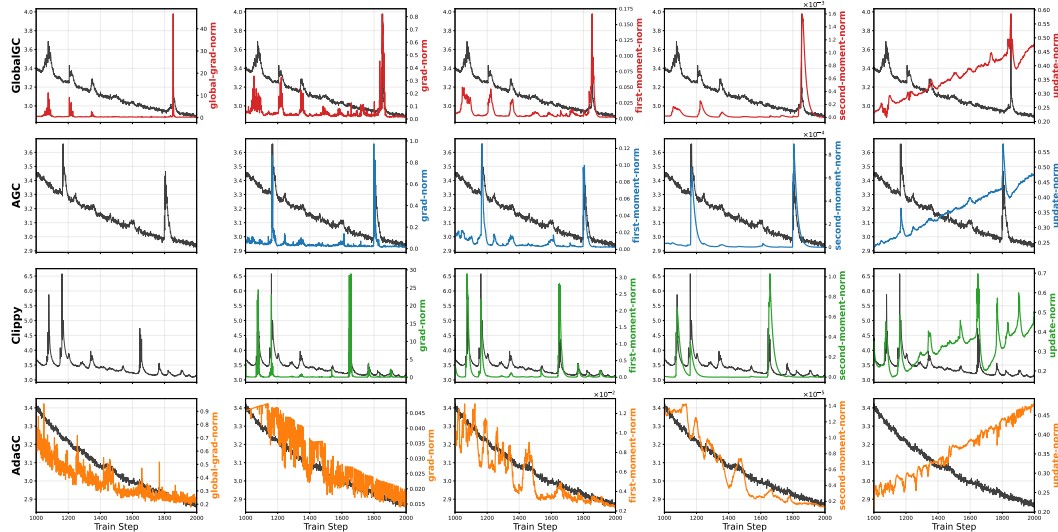

Figure 2: Visualization of the gradient norm, first-moment norm, second-moment norm, update norm, loss, and global gradient norm for the `embedding` of Llama-2 1.3B during warmup phase. Each row represents a different clipping method: the first row is GlobalGC, the second is AGC, the third is Clippy, and the fourth is our AdaGC. The black curve in each plot shows the loss trajectory.

# 4    METHODOLOGY: ADAGC

## 4.1    PRELIMINARIES

**Notations.** Let $x_t \in \mathbb{R}^d$ denote a parameter vector where $x_t^j$ represents its $j$-th coordinate for $j \in [d]$. We write $\nabla_x f(x)$ for the gradient of any differentiable function $f : \mathbb{R}^d \to \mathbb{R}$, and use $u^2$ and $u/v$ to denote element-wise square and division operations for vectors $u, v \in \mathbb{R}^d$. The $\ell_2$-norm and $\ell_\infty$-norm are denoted by $\|\cdot\|$ and $\|\cdot\|_\infty$, respectively. For asymptotic comparisons, we write $f = \mathcal{O}(g)$ if $\exists c > 0$ such that $f(x) \leq cg(x)$ for all $x$ in the domain.

**Gradient Clipping Fundamentals.** Consider a stochastic optimization problem with parameters $\theta \in \mathbb{R}^d$ and loss function $f(\theta; X_t)$ evaluated on mini-batch $X_t$ at step $t$. Standard gradient descent updates follow:

$$\theta_t = \theta_{t-1} - \eta_t \nabla_\theta f(\theta_{t-1}, X_t) \tag{1}$$

To prevent unstable updates from gradient explosions, GlobalGC (Pascanu et al., 2013) modifies the update rule as:

$$\theta_t = \theta_{t-1} - \eta_t h_t \nabla_\theta f(\theta_{t-1}, X_t)$$
$$\text{where } h_t := \min\left\{ \frac{\lambda_{abs}}{\|\nabla_\theta f(\theta_{t-1}; X_t)\|}, 1.0 \right\} \tag{2}$$

Here $\lambda_{abs}$ is an absolute clipping threshold requiring careful tuning, and $\eta_t$ is the learning rate. Our work focuses on *norm-based* clipping (scaling entire gradients exceeding $\lambda_{abs}$) rather than *value-based* clipping (element-wise truncation).

## 4.2    ADAPTIVE GRADIENT CLIPPING BASED ON LOCAL GRADIENT NORM

This section introduces a novel gradient clipping strategy termed AdaGC, which distinguishes itself by not relying on a global gradient norm. Instead, AdaGC focuses on the local gradient norm of each tensor and utilizes a dynamic adaptive mechanism for gradient clipping. The proposed method employs an EMA mechanism to maintain smoothed estimates of historical gradient norms per tensor, thus enhancing the accuracy of anomalous gradient detection and enabling independent clipping adjustments tailored to each tensor's specific conditions. EMA is widely used in deep learning, and within AdaGC, it facilitates the balancing of historical and current gradient norms. The formulation

is as follows:

$$\boldsymbol{g}_{t,i} \leftarrow h_{t,i} \cdot \boldsymbol{g}_{t,i}, \text{where } h_{t,i} = \min\left\{\lambda_{rel}\frac{\gamma_{t-1,i}}{\|\boldsymbol{g}_{t,i}\|}, 1.0\right\},$$

$$\gamma_{t,i} = \beta\gamma_{t-1,i} + (1-\beta)\|\boldsymbol{g}_{t,i}\|. \tag{3}$$

Here, $\lambda_{rel}$ is a predefined relative clipping threshold, $\boldsymbol{g}_{t,i}$ represents the gradient of the $i$-th tensor at time step $t$, and $h_{t,i}$ is a clipping function activated when $\|\boldsymbol{g}_{t,i}\| > \lambda_{rel} \cdot \gamma_{t-1,i}$, thereby scaling the gradient norm to $\lambda_{rel} \cdot \gamma_{t-1,i}$. Additionally, $\beta$ is the smoothing coefficient for EMA. *We consistently incorporate the clipped gradient norm into the historical observations rather than the pre-clipped values.*

Despite its simplicity, AdaGC adaptively adjusts based on the magnitude of each tensor's gradient norm. Whenever the gradient norm at a current timestep exceeds a predefined range of average norms within a historical window, it effectively suppresses these outlier gradients.

However, during the initial stages of model training (e.g., the first 100 steps), the gradient norms are typically large and fluctuate significantly, indicating a substantial decreasing trend. Direct application of AdaGC during this period could lead to two issues: first, erroneously accumulating the early large gradient norms into the historical values, resulting in compounded errors; second, compared to GlobalGC, AdaGC might delay clipping, thus potentially slowing down the loss reduction. To address these issues, we introduce a hyperparameter $T_{start}$ (default set to 100), representing a warm-up period during which traditional GlobalGC is applied.

Additionally, AdaGC is optimizer-agnostic, can be seamlessly integrated with various optimizers, such as AdamW (Loshchilov & Hutter, 2017), Lion (Chen et al., 2024), Muon (Jordan et al., 2024), enhancing its practicality and flexibility. Algorithm 1 in Appendix B demonstrates its implementation with the AdamW optimizer.

### 4.3 MEMORY, COMPUTATION, AND COMMUNICATION

**Memory.** As a tensor-wise method, AdaGC maintains an EMA of gradient norms for each parameter tensor, requiring storage of a single 32-bit float (4 bytes) per tensor. For ERNIE models, the total additional memory overhead has complexity of $\mathcal{O}((9 + 3E) \times L + 3)$, where $L$ and $E$ denote the number of transformer layers and experts, respectively. Specifically, this includes four tensors from the attention module per layer, $3 \times (1 + E)$ tensors from the shared and router experts per layer, and two RMSNorm tensors per layer; plus one tensor each for the embedding layer, the final layer normalization, and the language modeling head. In practice, this added memory footprint is negligible compared to the overall memory requirements of large-scale model training.

**Computation.** The computational cost of computing $\ell_2$ norms is the same for both AdaGC and GlobalGC. The difference is that GlobalGC applies a uniform scaling to all gradients, while AdaGC scales each gradient tensor independently.

**Communication.** In setups involving data parallelism (DP), tensor parallelism (TP), and pipeline parallelism (PP), GlobalGC requires an all-reduce operation across all DP, TP, and PP groups to aggregate the global norm. In contrast, AdaGC only needs an all-reduce within each TP group to compute per-tensor local norms. This design substantially reduces communication overhead, offering increasing benefits as model and cluster sizes grow.

### 4.4 CONVERGENCE ANALYSIS

Any operation that modifies gradients may potentially result in non-convergence. In this section, rather than providing a theoretical guarantee that AdaGC eliminates loss spikes, we present the convergence guarantee for Adam with AdaGC, stated as follows:

**Theorem 4.1** *Under mild assumptions, by selecting $\alpha_t = \mathcal{O}(1/\sqrt{T})$, $\beta_2 = 1 - \mathcal{O}(1/T)$ and $\beta_1 < \sqrt{\beta_2}$, when $\tau$ is randomly chosen from $\{1, 2, \cdots, T\}$ with equal probabilities, it holds that*

$$\mathbb{E}\|\nabla f(\theta_\tau)\|^2 = \mathcal{O}\left(\frac{1}{\sqrt{T}}\right).$$

Table 2: Zero-shot accuracy of AdaGC on Llama-2 7B under different hyperparameters.

| $\lambda_{rel}$ \ $\beta$ | 0.98 | 0.985 | 0.99 | 0.999 |
|---|---|---|---|---|
| 1.03 | 50.06 | 50.92 | 50.95 | 50.96 |
| 1.04 | 48.88 | 50.59 | **51.04** | 50.76 |
| 1.05 | 51.01 | 49.95 | 50.57 | 50.74 |

Table 3: Two-shot accuracy of AdaGC on Llama-2 7B under different hyperparameters.

| $\lambda_{rel}$ \ $\beta$ | 0.98 | 0.985 | 0.99 | 0.999 |
|---|---|---|---|---|
| 1.03 | 52.31 | 52.68 | 53.13 | 53.42 |
| 1.04 | 52.68 | 53.01 | **53.47** | 52.85 |
| 1.05 | 52.68 | 52.67 | 51.96 | 53.51 |

Theorem 4.1 shows that even with local clipped gradient, Adam with AdaGC can converge at the same rate as vanilla Adam (Kingma & Ba, 2014). Due to the limited space, the formal assumptions and theorem statement with detailed proof can be found in Appendix A.

## 5 EXPERIMENTS

### 5.1 EXPERIMENTAL SETUP

**Models and Datasets.** AdaGC is designed to enhance training stability during large language model pretraining. We evaluate its effectiveness on both dense and MoE (Mixture-of-Experts) architectures. For dense models, we use Llama-2 with 1.3B and 7B parameters. For MoE models, we experiment with Mixtral 8×1B (Jiang et al., 2024) and ERNIE 10B-A1.4B (Baidu-ERNIE-Team, 2025), where Mixtral 8×1B is a scaled-down version of Mixtral 8×7B, and ERNIE 10B-A1.4B is derived from ERNIE-4.5 21B-A3B. For pre-training, we use C4-en (Raffel et al., 2020), a clean English text corpus extracted from Common Crawl.

**Comparison Methods.** We focus on *clipping-based* methods and compare gradient and update clipping baselines, including GlobalGC (Pascanu et al., 2013), Gradient Value Clipping (ClipBy-Value), AGC (Brock et al., 2021), and Clippy (Tang et al., 2023). We also evaluate recent methods, including SPAM (Huang et al., 2025), Scaled Embed (Takase et al., 2023), and WeSaR (Nishida et al., 2024). Results are in Appendix E.2 Table 11.

**Training Details.** Pre-training large-scale models is typically resource-intensive. Our primary focus was to explore training instability rather than achieve ultimate accuracy. For ease of multiple experiments, we conducted 9,000 training steps on 36 billion tokens for both Llama-2 1.3B and 7B, 36,000 steps on 36 billion tokens for the Mixtral 8x1B, and 21,000 steps on 350 billion tokens for ERNIE 10B-A1.4B. We further trained ERNIE 10B–A1.4B for 60,000 steps on 1 trillion tokens to additionally validate the long-term stability of AdaGC. For additional details on the hyperparameters, please refer to Table 8 of Appendix C.

**Evaluation Metrics.** To quantitatively assess training stability, we follow (OLMo et al., 2024; Karpathy, 2024) and adopt the *spike score* as an objective metric. Specifically, the spike score is defined as the percentage of values in a time series that deviate by at least ten standard deviations from a rolling average of the preceding 1,000 values. This metric is primarily applied to training loss to detect sudden instabilities. Additionally, we evaluate performance using the training loss and validation perplexity (PPL) curves, as well as standard benchmark results, to provide a comprehensive assessment of convergence efficiency and model quality.

**Standard Benchmark.** We conducted a comprehensive evaluation of the model's zero-shot and two-shot capabilities across seven well-established benchmarks: ARC (Yadav et al., 2019), BoolQ (Clark et al., 2019), HellaSwag (Zellers et al., 2019), OBQA (Mihaylov et al., 2018), PIQA (Bisk et al., 2020), WinoGrande (Sakaguchi et al., 2021), and MMLU (Hendrycks et al., 2020). Following standard practice (Zhang et al., 2025), we report accuracy norm for ARC-E, ARC-C, HellaSwag, OBQA, and SciQ, as well as standard accuracy for all other tasks. For ERNIE 10B-A1.4B, which has been trained on 350B tokens, we evaluate its general abilities on a range of benchmarks, including MMLU (Hendrycks et al., 2020), GSM8K (Cobbe et al., 2021), BBH (Suzgun et al., 2022), TruthfulQA (Lin et al., 2021), and HumanEval (Chen et al., 2021). These benchmarks assess the model's enhanced capabilities in performing diverse downstream tasks, such as examination, reasoning, factuality, and coding.

## 5.2 CRITICAL HYPERPARAMETER SELECTION

We systematically evaluated two key hyperparameters in AdaGC: the EMA coefficient $\beta$ and the relative clipping threshold $\lambda_{rel}$. Specifically, we performed a grid search on the Llama-2 7B model to optimize these two hyperparameters, using zero-shot and two-shot performance across multiple tasks as evaluation metrics. As shown in Tables 2 and 3, the best performance was achieved when $\lambda_{rel} = 1.04$ and $\beta = 0.99$. We therefore adopted this configuration as the default setting for subsequent experiments and terminated further hyperparameter search. In addition, as observed in Tables 2 and 3, AdaGC's performance remains relatively stable across different hyperparameter values, suggesting that the method is robust to hyperparameter variations.

## 5.3 MAIN EXPERIMENTAL RESULTS

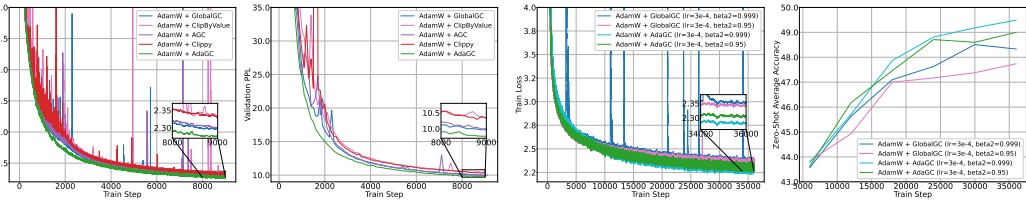

(a) Llama-2 7B training dynamics.  (b) Mixtral 8x1B training dynamics.

Figure 3: Large language model training analysis: Llama-2 7B and Mixtral 8x1B model comparison shows AdaGC's loss spike elimination and performance gains.

**Training Stability.** Our comprehensive evaluation shows AdaGC's effectiveness in improving training stability across a range of model scales and architectures. As shown in Figure 3, we compare the training dynamics of Llama-2 7B and Mixtral 8×1B models in terms of loss trajectories, validation perplexity, and zero-shot average accuracy. For the 7B models, baseline methods (GlobalGC, ClipByValue, AGC, Clippy) consistently exhibit frequent loss spikes during training, while AdaGC effectively eliminates these instability events. On Mixtral 8×1B, using the default $\beta_2 = 0.999$ leads to recurrent loss spikes, whereas decreasing $\beta_2$ to 0.95 helps mitigate this issue, indicating the strong impact of $\beta_2$ on training stability. AdaGC, however, can eliminate loss spikes for both $\beta_2 = 0.999$ and $\beta_2 = 0.95$, further demonstrating its robustness. The zero-shot average accuracy curves also reveal that AdaGC not only stabilizes training under $\beta_2 = 0.999$, but also improves convergence performance. For the ERNIE 10B-A1.4B, Figure 1b shows that stable convergence is achieved with $\epsilon = 1e-15$, which is particularly advantageous for large-scale models as it enables more parameters to fully utilize the adaptive learning rate in AdamW. Furthermore, Figure 2 illustrates AdaGC's clipping process, which prevents abnormal gradients from entering optimizer states, further smoothing parameter updates and reducing oscillations, thereby benefiting training stability.

**Spike Score Analysis.** Table 4 quantitatively summarizes the reduction in spike score achieved by AdaGC and the baseline methods across various settings. For Llama-2 7B, the spike score is reduced from 0.0333 with GlobalGC to 0 with AdaGC; for Mixtral 8×1B, it drops from 0.0144 to 0; and for ERNIE 10B-A1.4B, from 0.01 to 0. These results consistently demonstrate that AdaGC effectively and robustly eliminates loss spikes compared to existing clipping methods.

Table 4: Comparison of spike scores for various models under different clipping methods.

| Model | Llama-2 7B | | | | | Mixtral 8x1B | | ERNIE 10B-A1.4B | |
|---|---|---|---|---|---|---|---|---|---|
| Method | GlobalGC | ClipByValue | AGC | Clippy | **AdaGC** | GlobalGC | **AdaGC** | GlobalGC | **AdaGC** |
| Total Steps | 9K | 9K | 9K | 9K | 9K | 36K | 36K | 21K | 21K |
| Num Spikes | 3 | 9 | 8 | 3 | 0 | 52 | 0 | 2 | 0 |
| Spike Score (%) | 0.0333 | 0.1000 | 0.0889 | 0.0333 | **0.0000** | 0.0144 | **0.0000** | 0.0100 | **0.0000** |

**Results on Downstream Benchmarks.** Downstream zero-shot and two-shot evaluation results on the Llama-2 1.3B/7B and Mixtral 8×1B models (see Table 5 and Table 10) clearly demonstrate the

practical benefits of stable training. Across all model scales, AdaGC consistently achieves state-of-the-art performance or matches the best baselines. Specifically, on Llama-2 7B and Mixtral $8\times1$B, AdaGC obtains superior zero-shot (51.01% / 49.01%) and two-shot (53.47% / 51.61%) average accuracy, surpassing the GlobalGC baseline by +1.32% / +1.27% and +0.83% / +1.14%, respectively. Furthermore, long-term training of ERNIE 10B-A1.4B on 350B tokens shows that AdaGC achieves more stable convergence with $\epsilon = 1e - 15$, resulting in a 2.48% improvement over GlobalGC on the general abilities validation set. These findings establish a strong correlation between training stability and final model quality, indicating that the stability enabled by AdaGC facilitates better convergence and enhanced downstream performance.

Table 5: The Zero-Shot evaluation results of Llama-2 1.3B/7B and Mixtral 8x1B models on standard benchmarks.

| Model | Method | ARC-E acc_norm | ARC-C acc_norm | BoolQ acc | HellaSw. acc_norm | OBQA acc_norm | PIQA acc_norm | W.G. acc | MMLU acc | SciQ acc_norm | Avg. |
|---|---|---|---|---|---|---|---|---|---|---|---|
| Llama-2 1.3B | GlobalGC | **43.18** | **25.68** | 57.19 | 46.62 | 30.20 | **69.97** | 52.64 | 22.97 | 68.40 | 46.32 |
| | ClipByValue | 42.17 | **25.68** | **59.94** | 44.11 | **30.40** | 69.59 | 53.28 | **22.99** | 68.00 | 46.24 |
| | Clippy | 41.71 | 24.66 | 56.51 | 45.43 | 30.00 | 69.21 | **54.85** | 22.90 | 67.50 | 45.86 |
| | **AdaGC** | 42.09 | 25.51 | 58.01 | **47.29** | **30.40** | 69.70 | 52.33 | 22.98 | **68.70** | **46.33** |
| Llama-2 7B | GlobalGC | 49.49 | 27.56 | 56.30 | 56.06 | 33.60 | **74.59** | 55.33 | 23.12 | 71.20 | 49.69 |
| | ClipByValue | 46.21 | 26.88 | 57.03 | 53.49 | 33.20 | 71.65 | 53.59 | 23.36 | 70.50 | 48.43 |
| | AGC | 48.15 | 28.16 | 52.87 | 55.47 | **32.80** | 72.74 | 57.85 | **24.33** | 71.70 | 49.34 |
| | Clippy | 47.69 | 27.73 | **57.46** | 53.34 | 32.40 | 72.74 | 54.38 | 25.36 | 73.40 | 49.39 |
| | **AdaGC** | **49.58** | **28.92** | 57.28 | **57.94** | **32.80** | 74.32 | **58.09** | 23.62 | **76.60** | **51.01** |
| Mixtral 8x1B | GlobalGC | 44.70 | 25.94 | 56.57 | 53.08 | **33.00** | 71.60 | **54.70** | 22.91 | 67.20 | 47.74 |
| | **AdaGC** | **46.68** | **26.37** | **58.93** | **55.85** | 32.20 | **73.12** | 54.38 | **23.22** | **70.30** | **49.01** |

Table 6: Evaluation results of ERNIE 10B-A1.4B on multiple benchmarks after 21,000 (350B tokens) and 60,000 (1T tokens) training steps, comparing different optimization configurations.

| Steps (tokens) | Method | AdamW eps | MMLU | GSM8K | BBH | TruthfulQA | HumanEval | Avg. |
|---|---|---|---|---|---|---|---|---|
| 21k (350B) | GlobalGC | 1e-8 | 47.75 | **28.35** | 28.80 | 22.02 | 19.51 | 28.09 |
| | GlobalGC | 1e-15 | 39.11 | 21.46 | **29.35** | 23.39 | 15.24 | 25.71 |
| | AdaGC | 1e-15 | **42.07** | 25.32 | 27.89 | **24.92** | **20.73** | **28.19** |
| 60k (1T) | GlobalGC | 1e-8 | 48.61 | 39.88 | 30.84 | 30.73 | 22.56 | 34.52 |
| | GlobalGC | 1e-15 | 48.48 | **40.79** | 30.59 | 28.29 | **23.78** | 34.38 |
| | AdaGC | 1e-15 | **48.70** | 36.01 | **31.38** | **35.02** | 22.56 | **34.73** |

## 5.4 OPTIMIZER COMPATIBILITY: MUON AND LION

AdaGC is an optimizer-agnostic gradient clipping method that can be seamlessly integrated not only with AdamW, but also with other optimizers. To verify the generality of AdaGC, we conducted experiments on both LLM and VLM tasks by combining Llama-2 1.3B and CLIP ViT-Base models with the Muon and Lion optimizers, respectively, and compared them against GlobalGC. Although no loss spikes were observed under either experimental setting, AdaGC consistently demonstrated strong compatibility and generalization. In downstream zero-shot average accuracy, AdaGC outperformed GlobalGC by 0.14% (47.18% vs. 47.04%) with Muon and by 0.16% (40.81% vs. 40.65%) with Lion. These results further confirm that AdaGC can be effectively applied across different optimizers, providing stable training and improved downstream performance.

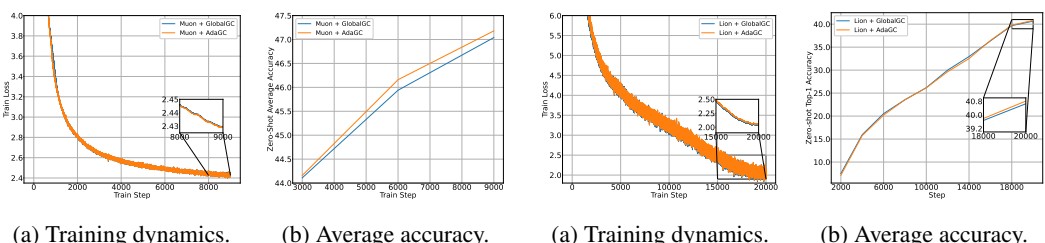

| (a) Training dynamics. | (b) Average accuracy. | (a) Training dynamics. | (b) Average accuracy. |
|---|---|---|---|

Figure 4: AdaGC with Muon on Llama-2 1.3B.    Figure 5: AdaGC with Lion on CLIP ViT-Base.

## 5.5 END-TO-END TRAINING WALL-CLOCK

Table 7 compares the GPU hours required for training various models using different distributed parallelism strategies. Compared to GlobalGC, AdaGC reduces end-to-end GPU hours by 0.27% on Llama-2 1.3B, 4.48% on Llama-2 7B, 1.24% on Mixtral 8x1B, and 1.53% on ERNIE 10B-A1.4B, thanks to reduced communication overhead. This highlights AdaGC's additional communication and efficiency benefits in large-scale distributed training.

Table 7: GPU hours under the same configuration. DPS denotes distributed parallel strategies.

| Model | Llama-2 1.3B | Llama-2 7B | Mixtral 8x1B | ERNIE 10B-A1.4B |
|---|---|---|---|---|
| DPS | DP=256, TP=1, PP=1 | DP=32, TP=2, PP=1 | DP=256, TP=1, PP=1, EP=1 | DP=64, TP=1, PP=4, EP=8 |
| Steps | 9K | 9K | 36K | 21K |
| GlobalGC | 513.0 | 1468.2 | 2060.8 | 22922 |
| **AdaGC** | 511.6 | 1402.4 | 2035.2 | 22572 |

## 5.6 ABLATION STUDY

We conduct systematic ablation studies across three critical dimensions of AdaGC: (1) EMA gradient norm initialization strategies, (2) GlobalGC warm-up steps, (3) adaptivity efficacy, and (4) locality granularity.

**EMA Initialization Strategy**. The initialization of EMA gradient norms requires careful design due to large initial gradient fluctuations during early training phases (first 100 steps). We evaluate five initialization variants: The default AdaGC strategy employs GlobalGC during warm-up while tracking minimum per-parameter norms ($\gamma_{t,i} = \min(\|\boldsymbol{g}_{t,i}\|, \gamma_{t-1,i})$). Comparative approaches include: (1) norm initialization

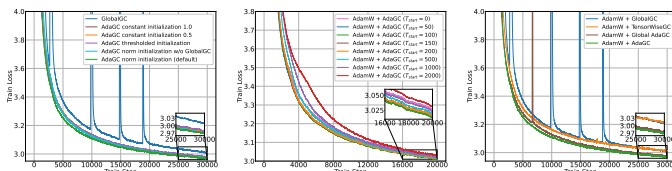

(a) $\gamma_{t,i}$ initialization.    (b) $T_{start}$ warm-up.    (c) Adaptivity, locality.

Figure 6: Training dynamics of ablation studies on AdaGC, showing (a) the influence of different EMA initialization strategies; (b) the impact of the GlobalGC warm-up steps $T_{start}$; and (c) the effects of adaptivity and locality granularity on gradient clipping efficacy and final loss.

without GlobalGC warm-up (directly using $\gamma_{t,i} = \min(\|\boldsymbol{g}_{t,i}\|, \gamma_{t-1,i})$ from step 0), (2) constant initialization ($\gamma_{0,i} \in \{0.5, 1.0\}$), and (3) thresholded initialization ($\gamma_{t,i} = \min(\|\boldsymbol{g}_{t,i}\|, 0.1)$). Figure 6a demonstrates that while all variants eliminate loss spikes, convergence quality varies within 0.36%. The default strategy achieves optimal final loss (2.9708 vs 2.9725 for next-best), showing that GlobalGC-guided warm-up better preserves parameter update consistency than direct initialization. This establishes the importance of phased initialization for gradient norm adaptation.

**Warm-up Steps** $T_{start}$. To further investigate whether the choice of GlobalGC warm-up steps $T_{start}$ has a significant impact and to provide practical guidance for practitioners, we additionally evaluate $T_{start} = \{0, 50, 100, 150, 200, 500, 1000, 2000\}$. The results in Figure 6b show that $T_{start} = 100$ consistently achieves the best performance. According to the EMA initialization formula $\gamma_{t,i} = \min(\|\boldsymbol{g}_{t,i}\|, \gamma_{t-1,i})$, an excessively large $T_{start}$ accumulates lower $\gamma_{t,i}$ values due to early training dynamics, which may lead to over-clipping and suppressed convergence in later training. Conversely, an overly small $T_{start}$ accumulates larger $\gamma_{t,i}$ values, which may delay clipping and hinder timely suppression of abnormal gradients. In contrast, $T_{start} = 100$ introduces negligible additional overhead for large-scale training while providing consistently stable performance improvements.

**Adaptivity Efficacy and Locality Granularity**. We conduct three sets of ablation experiments to evaluate the adaptivity and locality of AdaGC. The baseline uses GlobalGC (no adaptivity, no locality) with a fixed threshold of 1.0. In comparison, we examine (1) adaptive global gradient norm clipping (Global AdaGC, adaptive but non-local), which employs a single adaptive threshold for the entire model, (2) tensor-wise gradient norm clipping (TensorWiseGC, local but non-adaptive),

which allocated each tensor's fixed clipping threshold proportionally to its parameter count relative to the entire model, and (3) tensor-wise adaptation (AdaGC, adaptive and local), which adjusts thresholds independently for each tensor. As shown in Figure 6c, Global AdaGC reduces but does not completely eliminate spike events (1 event vs. 0 for tensor-wise) and yields a 0.25% higher final loss (2.9639 vs. 2.9712). Although TensorWiseGC also mitigates loss spikes, it noticeably slows down convergence and requires careful per-tensor threshold tuning to perform well. These results demonstrate that tensor-wise adaptive clipping provides both greater spike suppression and lower loss than other approaches.

## 6 CONCLUSION

The factors triggering loss spikes in large-scale pretraining are diverse and remain an open research problem, with no unified solution to date. Unlike prior work that seeks to identify root causes, we focus on a gradient-centric remedy and introduce AdaGC, an adaptive per-tensor gradient clipping method that prevents abnormal gradients from contaminating optimizer states. This approach ensures smoother updates and effectively eliminates loss spikes. Extensive experiments demonstrate that AdaGC delivers robust and stable training across both dense and MoE models, from 1.3B to 10B parameters, consistently reducing spike scores to zero and improving benchmark performance. Our results highlight AdaGC as a simple and effective solution for stable large-scale model pretraining.

## 7 STATEMENT ON THE USE OF LLMS

In preparing this manuscript, LLMs (mostly GPT-4.1/5) is utilized for linguistic refinement, including the detection and correction of grammar errors or spelling mistakes, and sentence rephrasing to improve clarity, coherence and readability. LLMs were also referenced when structuring the paper contents, and review missing details, but not involved in the formulation of ideas, the execution of experiments, or the generation of experimental results in this article.

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

# A CONVERGENCE PROOF

In this section, we provide the necessary assumptions and lemmas for the proofs of Theorem 4.1.

**Notations** The $k$-th component of a vector $v_t$ is denoted as $v_{t,k}$. Other than that, all computations that involve vectors shall be understood in the component-wise way. We say a vector $v_t \geq 0$ if every component of $v_t$ is non-negative, and $v_t \geq w_t$ if $v_{t,k} \geq w_{t,k}$ for all $k = 1, 2, \ldots, d$. The $\ell_1$ norm of a vector $v_t$ is defined as $\|v_t\|_1 = \sum_{k=1}^{d} |v_{t,k}|$. The $\ell_2$ norm is defined as $\|v_t\|^2 = \langle v_t, v_t \rangle = \sum_{k=1}^{d} |v_{t,k}|^2$. Given a positive vector $\hat{\eta}_t$, it will be helpful to define the following weighted norm: $\|v_t\|_{\eta_t}^2 = \langle v_t, \hat{\eta}_t v_t \rangle = \sum_{k=1}^{d} \hat{\eta}_{t,k} |v_{t,k}|^2$.

**Assumption A.1** *The function $f$ is lower bounded by $\underline{f}$ with L-Lipschitz gradient.*

**Assumption A.2** *The gradient estimator $g$ is unbiased with bounded norm, e.g,*
$$\mathbb{E}[g|x_t] = \nabla f(x_t), \ \|g_t\| \leq G.$$

**Assumption A.3** *The coefficient of clipping $h_{t,i}$ is lower bounded by some $h_0 > 0$.*

**Assumption A.4** *$\|g_t - \nabla f(x_t)\| \leq p \|\nabla f(x_t)\|$ holds for some $p < 1$ and for all $t$.*

**Remark A.5** *Assumption A.1 and Assumption A.2 are widely used in the proof of optimization algorithm with adaptive learning rates (Reddi et al., 2018). Assumption A.3 is because the gradient norm changes slowly when training the neural network, and the last assumption holds when the batch size is large enough.*

**Lemma A.6** *Let $\zeta := \beta_1^2 / \beta_2$. We have the following estimate*
$$m_t^2 \leq \frac{1}{(1-\zeta)(1-\beta_2)} v_t, \ \forall t. \tag{4}$$

**Proof:** By the iteration formula $m_t = \beta_1 m_{t-1} + (1-\beta_1)\hat{g}_t$ and $m_0 = 0$, we have
$$m = \sum_{i=1}^{t} \beta_1^{t-i}(1-\beta_1)\hat{g}_i.$$

Similarly, by $v_t = \beta_2 v_{t-1} + (1-\beta_2)\hat{g}_t^2$ and $v_0 = 0$, we have
$$v_t = \sum_{i=1}^{t} \beta_2^{t-i}(1-\beta_2)\hat{g}_i^2$$

It follows by arithmetic inequality that
$$m_t^2 = \left( \sum_{i=1}^{t} \frac{(1-\beta_1)\beta_1^{t-i}}{\sqrt{(1-\beta_2)\beta_2^{t-i}}} \sqrt{(1-\beta_2)\beta_2^{t-i}} \hat{g}_i \right)^2$$
$$\leq \left( \sum_{i=1}^{t} \frac{(1-\beta_1)^2 \beta_1^{2(t-i)}}{(1-\beta_2)\beta_2^{t-i}} \right) \left( \sum_{i=1}^{t} (1-\beta_2)\beta_2^{t-i} \hat{g}_i^2 \right) = \left( \sum_{i=1}^{t} \frac{(1-\beta_1)^2 \beta_1^{2(t-i)}}{(1-\beta_2)\beta_2^{t-i}} \right) v_t.$$

Further, we have
$$\sum_{i=1}^{t} \frac{(1-\beta_1)^2 \beta_1^{2(t-i)}}{(1-\beta_2)\beta_2^{t-i}} \leq \frac{1}{1-\beta_2} \sum_{i=1}^{t} \left( \frac{\beta_1^2}{\beta_2} \right)^{t-i} = \frac{1}{1-\beta_2} \sum_{k=0}^{t-1} \zeta^k \leq \frac{1}{(1-\zeta)(1-\beta_2)}.$$

The proof is completed. $\square$

**Lemma A.7** *The following estimate holds*
$$\sum_{t=1}^{T} \|\Delta_t\|^2 \leq \frac{\alpha^2 G^2}{\epsilon}$$

**Proof:** By using the definition of $m_t$, it holds $\|m_t\|^2 \leq G^2$.

Then, $\|\Delta_t\|^2 = \|\frac{\alpha_t m_t}{\sqrt{v_t}+\epsilon}\|^2 \leq \frac{G^2}{\epsilon}\alpha_t^2$ by using the definition of $\Delta_t$.

Therefore, $\sum_{t=1}^{T}\|\Delta_t\|^2 \leq \frac{G^2}{\epsilon}\sum_{t=1}^{T}\frac{\alpha^2}{T} = \frac{G^2\alpha^2}{\epsilon}$.

$\square$

**Lemma A.8** *With the Assumption A.3 and A.4, it holds that*

$$\mathbb{E}\langle \nabla f(\theta_t), \hat{\eta}_t\hat{g}_t\rangle \geq h_0\mathbb{E}\|\nabla f(\theta_t)\|_{\hat{\eta}_t}^2 .$$

**Proof:** According to Assumption A.4, it holds that

$$\langle \nabla_i f(\theta_t), g_{t,i}\rangle = -\frac{1}{2}\left(\|\nabla_i f(\theta_t) - g_{t,i}\|^2 - \|\nabla_i f(\theta_t)\|^2 - \|g_{t,i}\|^2\right)$$
$$\geq (1-p^2)\|\nabla_i f(\theta_t)\|^2 \geq 0.$$

Thus, it holds that

$$\mathbb{E}[\langle \nabla f(x_t), \hat{\eta}_t\hat{g}_t\rangle] = \mathbb{E}\left[\sum_i \langle \nabla_i f(\theta_t), h_{t,i}\hat{\eta}_{t,i}g_{t,i}\rangle\right]$$
$$\geq h_0\mathbb{E}\left[\sum_i \langle \nabla_i f(x_t), h_{t,i}\hat{\eta}_{t,i}g_{t,i}\rangle\right]$$
$$= h_0\mathbb{E}\langle \nabla f(\theta_t), \hat{\eta}_t g_t\rangle = h_0\mathbb{E}\|\nabla f(\theta_t)\|_{\hat{\eta}_t}^2.$$

$\square$

Let $\Delta_t := \theta_{t+1} - \theta_t = -\alpha_t m_t/(\sqrt{v_t}+\epsilon)$. Let $\hat{v}_t = \beta_2 v_{t-1} + (1-\beta_2)\delta_t^2$, where $\delta_t^2 = \mathbb{E}_t[\hat{g}_t^2]$ and let $\hat{\eta}_t = \alpha_t/\sqrt{\hat{v}_t + \epsilon}$.

**Lemma A.9** *Let $M_t = \mathbb{E}[\langle \nabla f(\theta_t), \Delta_t\rangle + L\|\Delta_t\|^2]$. Let $\alpha_t = \alpha/\sqrt{T}$ and $\beta_2 = 1 - \beta/T$. Then, for $T \geq 1$ we have*

$$\sum_{t=1}^{T}M_t \leq \frac{C_2}{1-\sqrt{\zeta}} + \frac{LG^2\alpha^2}{(1-\sqrt{\zeta})\epsilon} - \frac{(1-\beta_1)h_0}{2}\sum_{t=1}^{T}\mathbb{E}\|\nabla f(\theta_t)\|_{\hat{\eta}_t}^2, \tag{5}$$

*where $C_2 = \frac{5}{2(1-\beta_1)h_0}\left((1-\beta_1)^2\frac{4\alpha\beta G^4}{\epsilon^3} + \beta_1^2\alpha\beta\left(\frac{G^4}{\beta_2\epsilon^3} + \frac{(1+\epsilon)G^2}{(1-\zeta)\epsilon\beta_2} + \frac{G^4}{\beta_2}\right)\right).$*

**Proof:** To split $M_t$, firstly we introduce the following two equalities. Using the definitions of $v_t$ and $\hat{v}_t$, we obtain

$$\frac{(1-\beta_1)\alpha_t\hat{g}_t}{\sqrt{v_t}+\epsilon} = \frac{(1-\beta_1)\alpha_t\hat{g}_t}{\sqrt{\hat{v}_t}+\epsilon} + (1-\beta_1)\alpha_t\hat{g}_t\left(\frac{1}{\sqrt{v_t}+\epsilon} - \frac{1}{\sqrt{\hat{v}_t}+\epsilon}\right)$$
$$= (1-\beta_1)\hat{\eta}_t\hat{g}_t + (1-\beta_1)\alpha_t\hat{g}_t\frac{(1-\beta_2)(\sigma_t^2 - \hat{g}_t^2)}{(\sqrt{v_t}+\epsilon)(\sqrt{\hat{v}_t}+\epsilon)(\sqrt{v_t}+\sqrt{\hat{v}_t})}$$
$$= (1-\beta_1)\hat{\eta}_t\hat{g}_t + (1-\beta_1)\hat{\eta}_t\hat{g}_t\frac{(1-\beta_2)(\sigma_t^2 - \hat{g}_t^2)}{(\sqrt{v_t}+\epsilon)(\sqrt{v_t}+\sqrt{\hat{v}_t})}$$

In addition, we can obtain:

$$\beta_1 \alpha_t m_{t-1} \left( \frac{1}{\sqrt{\beta_2 v_{t-1}} + \sqrt{\beta_2}\epsilon} - \frac{1}{\sqrt{v_t} + \epsilon} \right)$$

$$= \beta_1 \alpha_t m_{t-1} \frac{(1 - \beta_2) \hat{g}_t^2}{\left(\sqrt{v_t} + \epsilon\right) \left(\sqrt{\beta_2 v_{t-1}} + \sqrt{\beta_2}\epsilon\right) \left(\sqrt{v_t} + \sqrt{\beta_2 v_{t-1}}\right)} + \beta_1 \alpha_t m_{t-1} \frac{\left(1 - \sqrt{\beta_2}\right)\epsilon}{\left(\sqrt{v_t} + \epsilon\right) \left(\sqrt{\beta_2 v_{t-1}} + \sqrt{\beta_2}\epsilon\right)}$$

$$= \beta_1 \alpha_t m_{t-1} \frac{(1 - \beta_2) \hat{g}_t^2}{\left(\sqrt{\hat{v}_t} + \epsilon\right) \left(\sqrt{\beta_2 v_{t-1}} + \sqrt{\beta_2}\epsilon\right) \left(\sqrt{v_t} + \sqrt{\beta_2 v_{t-1}}\right)}$$

$$\quad + \beta_1 \alpha_t m_{t-1} \frac{(1 - \beta_2) \hat{g}_t^2}{\left(\sqrt{\beta_2 v_{t-1}} + \sqrt{\beta_2}\epsilon\right) \left(\sqrt{v_t} + \sqrt{\beta_2 v_{t-1}}\right)} \left( \frac{1}{\sqrt{\hat{v}_t} + \epsilon} - \frac{1}{\sqrt{v_t} + \epsilon} \right)$$

$$\quad + \beta_1 \alpha_t m_{t-1} \frac{\left(1 - \sqrt{\beta_2}\right)\epsilon}{\left(\sqrt{\hat{v}_t} + \epsilon\right) \left(\sqrt{\beta_2 v_{t-1}} + \sqrt{\beta_2}\epsilon\right)} + \beta_1 \alpha_t m_{t-1} \frac{\left(1 - \sqrt{\beta_2}\right)\epsilon}{\sqrt{\beta_2 v_{t-1}} + \sqrt{\beta_2}\epsilon} \left( \frac{1}{\sqrt{\hat{v}_t} + \epsilon} - \frac{1}{\sqrt{v_t} + \epsilon} \right)$$

$$= \beta_1 m_{t-1} \hat{\eta}_t \frac{(1 - \beta_2) \hat{g}_t^2}{\left(\sqrt{\beta_2 v_{t-1}} + \sqrt{\beta_2}\epsilon\right) \left(\sqrt{v_t} + \sqrt{\beta_2 v_{t-1}}\right)}$$

$$\quad + \beta_1 \hat{\eta}_t m_{t-1} \frac{(1 - \beta_2)^2 \hat{g}_t^2 (\sigma_t^2 - \hat{g}_t^2)}{(\sqrt{v_t} + \epsilon)(\sqrt{v_t} + \sqrt{\hat{v}_t})(\sqrt{\beta_2 v_{t-1}} + \sqrt{\beta_2}\epsilon)(\sqrt{v_t} + \sqrt{\beta_2 v_{t-1}})}$$

$$\quad + \beta_1 \hat{\eta}_t m_{t-1} \frac{(1 - \sqrt{\beta_2})\epsilon}{\sqrt{\beta_2 v_{t-1}} + \sqrt{\beta_2}\epsilon} + \beta_1 \hat{\eta}_t m_{t-1} \frac{(1 - \sqrt{\beta_2})(1 - \beta_2)\epsilon(\sigma_t^2 - \hat{g}_t^2)}{(\sqrt{v_t} + \epsilon)(\sqrt{v_t} + \sqrt{\hat{v}_t})(\sqrt{\beta_2 v_{t-1}} + \sqrt{\beta_2}\epsilon)}.$$

For simplicity, we denote

$$A_t^1 = (1 - \beta_1) \sqrt{\hat{\eta}_t} \hat{g}_t \frac{(1 - \beta_2) \left(\sigma_t^2 - \hat{g}_t^2\right)}{\left(\sqrt{v_t} + \epsilon\right) \left(\sqrt{v_t} + \sqrt{\hat{v}_t}\right)}$$

$$A_t^2 = \beta_1 m_{t-1} \sqrt{\hat{\eta}_t} \frac{(1 - \beta_2) \hat{g}_t^2}{\left(\sqrt{\beta_2 v_{t-1}} + \sqrt{\beta_2}\epsilon\right) \left(\sqrt{v_t} + \sqrt{\beta_2 v_{t-1}}\right)}$$

$$A_t^3 = \beta_1 \sqrt{\hat{\eta}_t} m_{t-1} \frac{(1 - \beta_2)^2 \hat{g}_t^2 (\sigma_t^2 - \hat{g}_t^2)}{(\sqrt{v_t} + \epsilon)(\sqrt{v_t} + \sqrt{\hat{v}_t})(\sqrt{\beta_2 v_{t-1}} + \sqrt{\beta_2}\epsilon)(\sqrt{v_t} + \sqrt{\beta_2 v_{t-1}})}$$

$$A_t^4 = \beta_1 \sqrt{\hat{\eta}_t} m_{t-1} \frac{(1 - \sqrt{\beta_2})\epsilon}{\sqrt{\beta_2 v_{t-1}} + \sqrt{\beta_2}\epsilon}$$

$$A_t^5 = \beta_1 \sqrt{\hat{\eta}_t} m_{t-1} \frac{(1 - \sqrt{\beta_2})(1 - \beta_2)\epsilon(\sigma_t^2 - \hat{g}_t^2)}{(\sqrt{v_t} + \epsilon)(\sqrt{v_t} + \sqrt{\hat{v}_t})(\sqrt{\beta_2 v_{t-1}} + \sqrt{\beta_2}\epsilon)}$$

Then, we obtain

$$\Delta_t - \frac{\beta_1 \alpha_t}{\sqrt{\beta_2} \alpha_{t-1}} \Delta_{t-1} = -\frac{\alpha_t m_t}{\sqrt{v_t} + \epsilon} + \frac{\beta_1 \alpha_t m_{t-1}}{\sqrt{\beta_2 v_{t-1}} + \sqrt{\beta_2}\epsilon}$$

$$= -\frac{(1 - \beta_1)\alpha_t \hat{g}_t}{\sqrt{v_t} + \epsilon} + \beta_1 \alpha_t m_{t-1} \left( \frac{1}{\sqrt{\beta_2 v_{t-1}} + \sqrt{\beta_2}\epsilon} - \frac{1}{\sqrt{v_t} + \epsilon} \right)$$

$$= -(1 - \beta_1)\hat{\eta}_t \hat{g}_t - \sqrt{\hat{\eta}_t} A_t^1 + \sqrt{\hat{\eta}_t} A_t^2 + \sqrt{\hat{\eta}_t} A_t^3 + \sqrt{\hat{\eta}_t} A_t^4 + \sqrt{\hat{\eta}_t} A_t^5$$

Thus, it holds that

$$\mathbb{E}\langle \nabla f(\theta_t), \Delta_t \rangle = \frac{\beta_1 \alpha_t}{\sqrt{\beta_2} \alpha_{t-1}} \langle \nabla f(\theta_t), \Delta_{t-1} \rangle + \mathbb{E}\left\langle \nabla f(\theta_t), \Delta_t - \frac{\beta_1 \alpha_t}{\sqrt{\beta_2} \alpha_{t-1}} \Delta_{t-1} \right\rangle$$

$$= \frac{\beta_1 \alpha_t}{\sqrt{\beta_2} \alpha_{t-1}} \left( \mathbb{E}\langle \nabla f(\theta_t), \Delta_{t-1} \rangle + \mathbb{E}\langle \nabla f(\theta_t) - \nabla f(\theta_{t-1}), \Delta_{t-1} \rangle \right)$$

$$\quad + \mathbb{E}\langle \nabla f(\theta_t), -(1 - \beta_1)\hat{\eta}_t \hat{g}_t \rangle + \mathbb{E}\langle \nabla f(\theta_t), -\sqrt{\hat{\eta}_t} A_t^1 \rangle + \mathbb{E}\langle \nabla f(\theta_t), \sqrt{\hat{\eta}_t} A_t^2 \rangle$$

$$\quad + \mathbb{E}\langle \nabla f(\theta_t), \sqrt{\hat{\eta}_t} A_t^3 \rangle + \mathbb{E}\langle \nabla f(\theta_t), \sqrt{\hat{\eta}_t} A_t^4 \rangle + \mathbb{E}\langle \nabla f(\theta_t), \sqrt{\hat{\eta}_t} A_t^5 \rangle$$

$$(6)$$

For the first term of equation 6, it holds that

$$\frac{\beta_1 \alpha_t}{\sqrt{\beta_2} \alpha_{t-1}} \left( \mathbb{E}\langle \nabla f(\theta_t), \Delta_{t-1} \rangle + \mathbb{E}\langle \nabla f(\theta_t) - \nabla f(\theta_{t-1}), \Delta_{t-1} \rangle \right)$$

$$\leq \frac{\beta_1 \alpha_t}{\sqrt{\beta_2} \alpha_{t-1}} \left( \mathbb{E}\langle \nabla f(\theta_t), \Delta_{t-1} \rangle + \mathbb{E}\|\nabla f(\theta_t) - \nabla f(\theta_{t-1})\|\|\Delta_{t-1}\| \right)$$

$$\leq \frac{\beta_1 \alpha_t}{\sqrt{\beta_2} \alpha_{t-1}} \left( \mathbb{E}\langle \nabla f(\theta_t), \Delta_{t-1} \rangle + L\mathbb{E}\|\Delta_{t-1}\|^2 \right)$$

$$= \frac{\beta_1 \alpha_t}{\sqrt{\beta_2} \alpha_{t-1}} M_{t-1}$$

For the second term of equation 6, it holds that

$$\mathbb{E}\langle \nabla f(\theta_t), -(1-\beta_1)\hat{\eta}_t \hat{g}_t \rangle \leq -(1-\beta_1)h_0 \mathbb{E}\|\nabla f(\theta_t)\|_{\hat{\eta}_t}^2.$$

For the rest of the terms, it holds that

$$\mathbb{E}\langle \nabla f(\theta_t), -\sqrt{\hat{\eta}_t} A_t^1 \rangle \leq \frac{h_0(1-\beta_1)}{10}\mathbb{E}\|\nabla f(\theta_t)\|_{\hat{\eta}_t}^2 + \frac{5}{2(1-\beta_1)h_0}\left\|A_t^1\right\|^2$$

$$\mathbb{E}\langle \nabla f(\theta_t), +\sqrt{\hat{\eta}_t} A_t^2 \rangle \leq \frac{h_0(1-\beta_1)}{10}\mathbb{E}\|\nabla f(\theta_t)\|_{\hat{\eta}_t}^2 + \frac{5}{2(1-\beta_1)h_0}\left\|A_t^2\right\|^2$$

$$\mathbb{E}\langle \nabla f(\theta_t), +\sqrt{\hat{\eta}_t} A_t^3 \rangle \leq \frac{h_0(1-\beta_1)}{10}\mathbb{E}\|\nabla f(\theta_t)\|_{\hat{\eta}_t}^2 + \frac{5}{2(1-\beta_1)h_0}\left\|A_t^3\right\|^2$$

$$\mathbb{E}\langle \nabla f(\theta_t), +\sqrt{\hat{\eta}_t} A_t^4 \rangle \leq \frac{h_0(1-\beta_1)}{10}\mathbb{E}\|\nabla f(\theta_t)\|_{\hat{\eta}_t}^2 + \frac{5}{2(1-\beta_1)h_0}\left\|A_t^4\right\|^2$$

$$\mathbb{E}\langle \nabla f(\theta_t), +\sqrt{\hat{\eta}_t} A_t^5 \rangle \leq \frac{h_0(1-\beta_1)}{10}\mathbb{E}\|\nabla f(\theta_t)\|_{\hat{\eta}_t}^2 + \frac{5}{2(1-\beta_1)h_0}\left\|A_t^5\right\|^2$$

On the other hand, it holds that

$$\left\|A_t^1\right\|^2 \leq (1-\beta_1)^2 \frac{4\alpha\beta G^4}{T\epsilon^3}, \left\|A_t^2\right\|^2 \leq \beta_1^2 \frac{\alpha\beta G^4}{T\beta_2\epsilon^3}, \left\|A_t^3\right\|^2 \leq \beta_1^2 \frac{\alpha\beta G^2}{(1-\zeta)\epsilon T\beta_2},$$

$$\left\|A_t^4\right\|^2 \leq \beta_1^2 \frac{\alpha\beta G^4}{T\beta_2}, \left\|A_t^5\right\|^2 \leq \beta_1^2 \frac{\alpha\beta G^2}{(1-\zeta)\beta_2 T}$$

$$\square$$

Define $N_t = \frac{C_2}{T} + L\mathbb{E}\|\Delta_t\|^2$, where $C_2 = \frac{5}{2(1-\beta_1)h_0}\left((1-\beta_1)^2\frac{4\alpha\beta G^4}{\epsilon^3} + \beta_1^2\alpha\beta\left(\frac{G^4}{\beta_2\epsilon^3} + \frac{(1+\epsilon)G^2}{(1-\zeta)\epsilon\beta_2} + \frac{G^4}{\beta_2}\right)\right)$.
It holds that

$$M_t \leq \frac{\beta_1 \alpha_t}{\sqrt{\beta_2}\alpha_{t-1}}M_{t-1} + N_t - \frac{1-\beta_1}{2}\hat{\eta}_t\mathbb{E}\|\nabla f(\theta_t)\|_{\hat{\eta}_t}^2 \leq \sum_{i=1}^t \sqrt{\zeta}^{t-i}N_i - \frac{1-\beta_1}{2}h_0\mathbb{E}\|\nabla f(\theta_t)\|_{\hat{\eta}_t}^2$$

Thus, by summing $t$ from 1 to $T$, it holds that

$$\sum_{t=1}^T M_t \leq \sum_{t=1}^T \sum_{i=1}^t \sqrt{\zeta}^{t-i}N_i - \frac{(1-\beta_1)h_0}{2}\mathbb{E}\|\nabla f(\theta_t)\|_{\hat{\eta}_t}^2$$

$$\leq \frac{1}{1-\sqrt{\zeta}}\sum_{t=1}^T N_t - \frac{(1-\beta_1)h_0}{2}\mathbb{E}\|\nabla f(\theta_t)\|_{\hat{\eta}_t}^2$$

$$\leq \frac{C_2}{1-\sqrt{\zeta}} + \frac{LG^2\alpha^2}{(1-\sqrt{\zeta})\epsilon} - \frac{(1-\beta_1)h_0}{2}\sum_{t=1}^T \mathbb{E}\|\nabla f(\theta_t)\|_{\hat{\eta}_t}^2.$$

**Lemma A.10** *Let $\tau$ be randomly chosen from $\{1, 2, \cdots, T\}$ with equal probabilities $p_\tau = \frac{1}{T}$. We have the following estimate:*

$$\mathbb{E}[\|\nabla f(\theta_\tau)\|^2] \leq \frac{\sqrt{G^2 + \epsilon d}}{\alpha\sqrt{T}}\mathbb{E}\left[\sum_{t=1}^T \|\nabla f(\theta_t)\|_{\hat{\eta}_t}^2\right].$$

**Proof:** Note that $\|\hat{v}_t\|_1 = \beta_2 \|v_{t-1}\|_1 + (1 - \beta_2) \|\sigma_t\|^2$ and $\|\hat{g}_t\| \leq G$. It is straightforward to prove $\|v_t\|_1 \leq G^2$. Hence, we have $\|\hat{v}_t + \epsilon\|_1 \leq G^2 + \epsilon d$.

Utilizing this inequality, we have

$$\|\nabla f(\theta_t)\|^2 = \frac{\|\nabla f(\theta_t)\|^2}{\sqrt{\|\hat{v}_t + \epsilon\|_1}} \sqrt{\|\hat{v}_t + \epsilon\|_1} = \sqrt{\|\hat{v}_t + \epsilon\|_1} \sum_{k=1}^d \frac{|\nabla_k f(\theta_t)|^2}{\sqrt{\sum_{l=1}^d \hat{v}_{t,l} + \epsilon}}$$

$$\leq \sqrt{\|\hat{v}_t + \epsilon\|_1} \alpha_t^{-1} \sum_{k=1}^d \frac{\alpha_t}{\sqrt{\hat{v}_{t,k} + \epsilon}} |\nabla_k f(\theta_t)|^2 = \sqrt{\|\hat{v}_t + \epsilon\|_1} \alpha_t^{-1} \|\nabla f(\theta_t)\|_{\hat{\eta}_t}^2$$

$$\leq \sqrt{G^2 + \epsilon d} \alpha_t^{-1} \|\nabla f(\theta_t)\|_{\hat{\eta}_t}^2 \leq \frac{\sqrt{G^2 + \epsilon d}}{\alpha_T} \|\nabla f(\theta_t)\|_{\hat{\eta}_t}^2.$$

Then, by using the definition of $\theta_\tau$, we obtain

$$\mathbb{E}\left[\|\nabla f(\theta_\tau)\|^2\right] = \frac{1}{T} \sum_{t=1}^T \mathbb{E}\left[\|\nabla f(\theta_t)\|^2\right] \leq \frac{\sqrt{G^2 + \epsilon d}}{\alpha \sqrt{T}} \mathbb{E}\left[\sum_{t=1}^T \|\nabla f(\theta_t)\|_{\hat{\eta}_t}^2\right].$$

Thus, the desired result is obtained. $\qquad\square$

**Theorem A.11** *Let $\{\theta_t\}$ be a sequence generated by AdaGC for initial values $\theta_1$ and $m_0 = v_0 = 0$. Assumptions A.1 to A.4 hold. With the hyperparameters $\alpha_t = \alpha/\sqrt{T}, \beta_2 = 1 - \beta/T$ and $\zeta = \beta_1^2/\beta_2 < 1$. Let $\tau$ be randomly chosen from $\{1, 2, \cdots, T\}$ with equal probabilities. We have*

$$\mathbb{E}\|\nabla f(\theta_\tau)\|^2 \leq \frac{C}{\sqrt{T}}$$

*where* $C = \frac{\sqrt{G^2 + \epsilon d}}{\alpha} \left(f(\theta_1) - \underline{f} + \frac{C_2}{1 - \sqrt{\zeta}} + \frac{LG^2\alpha^2}{(1 - \sqrt{\zeta})\epsilon}\right)$ *and* $C_2 = \frac{5}{2(1-\beta_1)h_0} \left((1 - \beta_1)^2 \frac{4\alpha\beta G^4}{\epsilon^3} + \beta_1^2 \alpha\beta \left(\frac{G^4}{\beta_2 \epsilon^3} + \frac{(1+\epsilon)G^2}{(1-\zeta)\epsilon\beta_2} + \frac{G^4}{\beta_2}\right)\right).$

**Proof:** With the Lipschitz continuity condition of $f$, it holds that

$$\mathbb{E}f(\theta_{t+1}) \leq \mathbb{E}\left[f(\theta_t) + \langle \nabla f(\theta_t), \Delta_t \rangle + \frac{L}{2}\|\Delta_t\|^2\right] \leq \mathbb{E}f(\theta_t) + M_t.$$

By summing $t$ from 1 to $T$, it holds that

$$\mathbb{E}f(\theta_{T+1}) \leq f(\theta_1) + \sum_{t=1}^T M_t \leq f(\theta_1) + \frac{C_2}{1 - \sqrt{\zeta}} + \frac{LG^2\alpha^2}{(1 - \sqrt{\zeta})\epsilon} - \frac{(1 - \beta_1)h_0}{2} \sum_{t=1}^T \mathbb{E}\|\nabla f(\theta_t)\|_{\hat{\eta}_t}^2$$

Thus, it holds that

$$\mathbb{E}\left[\|\nabla f(\theta_\tau)\|^2\right] \leq \frac{\sqrt{G^2 + \epsilon d}}{\alpha \sqrt{T}} \mathbb{E}\left[\sum_{t=1}^T \|\nabla f(\theta_t)\|_{\hat{\eta}_t}^2\right]$$

$$\leq \frac{\sqrt{G^2 + \epsilon d}}{\alpha \sqrt{T}} \left(f(\theta_1) - \mathbb{E}[f(\theta_{T+1})] + \frac{C_2}{1 - \sqrt{\zeta}} + \frac{LG^2\alpha^2}{(1 - \sqrt{\zeta})\epsilon}\right)$$

$$\leq \frac{\sqrt{G^2 + \epsilon d}}{\alpha \sqrt{T}} \left(f(\theta_1) - \underline{f} + \frac{C_2}{1 - \sqrt{\zeta}} + \frac{LG^2\alpha^2}{(1 - \sqrt{\zeta})\epsilon}\right)$$

$$\square$$

## B    Pseudocode of AdamW with AdaGC

Algorithm 1 presents the pseudocode of AdamW integrated with AdaGC. For clearer exposition, we highlight different components according to their origins: **orange** indicates the procedures inherited from the original GlobalGC algorithm, while **blue** is used to denote the new contributions and modifications introduced by AdaGC. Specifically, the GlobalGC steps include the global gradient clipping implemented via the scaling factor and the use of the clipped gradient in subsequent moments. The AdaGC components mainly comprise adaptive per-parameter clipping, the initialization and update of the adaptive threshold $\gamma_{t,i}$, and the warm-up strategy governed by $T_{start}$.

---

**Algorithm 1:** AdamW with AdaGC

---

1: **given:** $\{\eta_t\}_{t=1}^{T}, \lambda_w, \epsilon, \beta_1, \beta_2, \beta \in [0,1), \lambda_{abs}, T_{start}$
2: **initialize:** $\boldsymbol{\theta}_0, m_0 \leftarrow 0, v_0 \leftarrow 0, t \leftarrow 0$
3: **repeat**
4:     **compute** $\boldsymbol{g}_t = \nabla_{\boldsymbol{\theta}} f_t(\boldsymbol{\theta}_{t-1}, X_t)$
5:     **if** $t < T_{start}$ **then**
6:         $h_t = \min\left\{\frac{\lambda_{abs}}{\|\boldsymbol{g}_t\|}, 1.0\right\}$
7:         $\widehat{\boldsymbol{g}}_t = h_t \cdot \boldsymbol{g}_t$
8:         **for** $i \in |\boldsymbol{\theta}|$ **do**
9:             $\gamma_{t,i} = \min\left\{\gamma_{t-1,i}, \|\widehat{\boldsymbol{g}_{t,i}}\|\right\}, \gamma_{0,i} = \|\widehat{\boldsymbol{g}_{1,i}}\|$
10:        **end for**
11:    **else**
12:        **for** $i \in |\boldsymbol{\theta}|$ **do**
13:            $h_{t,i} = \min\left\{\lambda_{rel}\frac{\gamma_{t-1,i}}{\|\boldsymbol{g}_{t,i}\|}, 1.0\right\}$
14:            $\widehat{\boldsymbol{g}_{t,i}} = h_{t,i} \cdot \boldsymbol{g}_{t,i}$
15:            $\gamma_{t,i} = \beta\gamma_{t-1,i} + (1-\beta)\|\widehat{\boldsymbol{g}_{t,i}}\|$
16:        **end for**
17:    **end if**
18:    $\boldsymbol{m}_t = \beta_1\boldsymbol{m}_{t-1} + (1-\beta_1)\widehat{\boldsymbol{g}}_t$
19:    $\boldsymbol{v}_t = \beta_2\boldsymbol{v}_{t-1} + (1-\beta_2)\widehat{\boldsymbol{g}}_t^2$
20:    $\widehat{\boldsymbol{m}}_t = \boldsymbol{m}_t/(1-\beta_1^t), \quad \widehat{\boldsymbol{v}}_t = \boldsymbol{v}_t/(1-\beta_2^t)$
21:    $\boldsymbol{\theta}_t = \boldsymbol{\theta}_{t-1} - \eta_t\lambda_w\boldsymbol{\theta}_{t-1} - \eta_t\widehat{\boldsymbol{m}}_t/(\sqrt{\widehat{\boldsymbol{v}}_t} + \epsilon)$
22: **until** $\boldsymbol{\theta}_t$ not converge

---

## C    Hyper-Parameters

### C.1    Model Hyper-Parameters

Table 8 summarizes the model hyper-parameters used for all experiments. For each model, we report the core architecture settings (such as number of layers, hidden dimension, attention heads, and feedforward dimension), MoE-related configurations, and main optimization hyper-parameters (including learning rate, warmup, weight decay, and Adam parameters). Clipping thresholds $\lambda_{abs}$, $\lambda_{rel}$, and momentum $\beta$ are also listed, in correspondence with the techniques discussed in the main text. All experiments use a batch size and sequence length as shown, and we employ bfloat16 precision for most models except ERNIE, which uses float8. The symbol '–' indicates settings not applicable to a specific architecture.

### C.2    Clipping Hyper-Parameters

For other clipping methods, we primarily followed the recommended default settings from prior work, and performed limited tuning only when necessary to ensure a fair comparison.

Specifically:

- **GlobalGC**: We used the commonly adopted global clipping threshold $\lambda_{abs} = 1.0$ in large-scale pretraining.

Table 8: Hyper-parameters used in our LLMs experiments. $\lambda_{abs}$ represents the absolute global clipping threshold of GlobalGC. $\lambda_{rel}$ and $\beta$ represent the relative clipping threshold and the momentum of our AdaGC, respectively. The symbol '–' indicates that the parameter is not applicable.

| Model | LLaMA-1.3B | LLaMA-7B | ERNIE 10B-A1.4B | Mixtral 8x1B |
|---|---|---|---|---|
| Precision | bfloat16 | bfloat16 | float8 | bfloat16 |
| Num layers | 24 | 32 | 25 | 24 |
| Hidden dim size | 2048 | 4096 | 2560 | 2048 |
| FFN dim size | 5461 | 11008 | 1024 | 5632 |
| Num attention heads | 32 | 32 | 20 | 32 |
| Num key value heads | 32 | 32 | 4 | 4 |
| Attention bias | ✗ | ✗ | ✗ | ✗ |
| Num shared experts | - | - | 1 | 0 |
| Num router experts | - | - | 48 | 8 |
| Num experts per token | - | - | 3 | 2 |
| Sequence length | 2048 | 2048 | 4096 | 2048 |
| Batch size | 2048 | 2048 | 4096 | 512 |
| Iterations | 9000 | 9000 | 21000 | 36000 |
| Learning rate | $3.0 \times 10^{-4}$ | $3.0 \times 10^{-4}$ | $3.0 \times 10^{-4}$ | $3.0 \times 10^{-4}$ |
| LR decay | cosine | cosine | wsd | cosine |
| Warmup iterations | 2000 | 2000 | 2000 | 500 |
| Weight decay | 0.1 | 0.1 | 0.1 | 0.1 |
| Adam $\beta_1$ | 0.90 | 0.90 | 0.90 | 0.90 |
| Adam $\beta_2$ | 0.95 | 0.95 | 0.95 | 0.999 |
| $\lambda_{abs}$ | 1.0 | 1.0 | 1.0 | 1.0 |
| $\lambda_{rel}$ | 1.04 | 1.04 | 1.04 | 1.04 |
| $\beta$ | 0.99 | 0.99 | 0.99 | 0.99 |

- **ClipByValue**: Following the SPAM (Huang et al., 2025) setting, we set the clipping threshold to $\lambda_{abs} = 1e - 3$.

- **AGC**: We performed small-range tuning over $\lambda_{rel} \in \{1e - 2, 1e - 3, 1e - 4\}$ to find the best setting.

- **Clippy**: We tuned over $\lambda_{abs} \in \{0.1, 0.3, 0.5\}$ and $\lambda_{rel} \in \{1e - 2, 1e - 3, 1e - 4\}$ to select the optimal combination.

- **SPAM**: We adopted the default hyperparameters recommended for standard pretraining in the original paper, which were reported to perform well across diverse settings. Specifically, we set the interval to $\Delta T = 500$, the threshold to $\theta = 5000$, and the warmup steps to $N = 150$.

The final hyper-parameters used for other clipping methods are summarized in Table 9.

Table 9: Hyper-parameters for other clipping methods.

| Method | Hyperparameters |
|---|---|
| GlobalGC | $\lambda_{abs} = 1.0$ |
| ClipByValue | $\lambda_{abs} = 1e - 3$ |
| AGC | $\lambda_{rel} = 1e - 3$ |
| Clippy | $\lambda_{rel} = 1e - 3$ |
| SPAM | $\Delta T = 500, \theta = 5000, N = 150$ |

## D    EXPERIMENTAL DETAILS FOR CLIP

To further investigate the optimizer compatibility of AdaGC, we evaluated its effect on large-scale vision-language model pre-training, focusing on the CLIP ViT-Base model (Radford et al., 2021) with the Lion optimizer (Chen et al., 2024). The model comprises 151 million parameters and is trained on the LAION-400M (Schuhmann et al., 2021) dataset. Training is conducted for 20,000 steps, covering 320M image-text pairs.

The key training hyper-parameters are as follows: a learning rate of 0.002, weight decay of 0.2, and batch size of 32,768. We employ patch-dropout with a drop rate of 0.5 (Li et al., 2023), following recent best practices (Wortsman et al., 2023). The learning rate is linearly warmed up for the first 5,000 steps (Goyal et al., 2017), and subsequently decayed according to a cosine schedule (Loshchilov & Hutter, 2016).

Following pre-training, we report downstream zero-shot evaluation results on the ImageNet (Russakovsky et al., 2015) validation set. The results are shown in Figure 5 in the main text.

## E    MORE EVALUATION RESULTS

### E.1    RESULTS ON DOWNSTREAM BENCHMARKS

The Two-Shot evaluation results of Llama-2 1.3B/7B and Mixtral 8x1B models on standard benchmarks are presented in Table 10.

Table 10: The Two-Shot evaluation results of Llama-2 1.3B/7B and Mixtral 8x1B models on standard benchmarks. The best scores in each column are **bolded**. HellaSw. = HellaSwag, W.G. = WinoGrande.

| Model | Method | ARC-E acc_norm | ARC-C acc_norm | BoolQ acc | HellaSw. acc_norm | OBQA acc_norm | PIQA acc_norm | W.G. acc | MMLU acc | SciQ acc_norm | Avg. |
|---|---|---|---|---|---|---|---|---|---|---|---|
| Llama-2 1.3B | GlobalGC | **47.26** | 25.60 | **50.31** | 46.44 | **32.20** | 69.64 | 52.33 | 25.07 | 77.80 | 47.41 |
| | ClipByValue | 47.10 | 25.77 | 56.54 | 43.97 | 30.00 | 68.88 | 52.96 | **26.09** | 77.20 | **47.61** |
| | Clippy | 46.55 | 25.85 | 49.76 | 45.71 | 30.00 | **70.02** | 53.20 | 25.69 | 77.70 | 47.16 |
| | **AdaGC** | 46.04 | **26.19** | 49.72 | **47.51** | 31.00 | 69.70 | **54.38** | 24.98 | **78.50** | 47.56 |
| Llama-2 7B | GlobalGC | 55.81 | 28.58 | **60.70** | 56.54 | 33.00 | 73.72 | 56.75 | 25.51 | 83.20 | 52.64 |
| | ClipByValue | 51.94 | 26.88 | 57.55 | 53.36 | 32.40 | 72.31 | 54.14 | 26.63 | 81.60 | 50.75 |
| | AGC | 52.95 | 28.67 | 56.15 | 55.69 | **35.40** | 73.07 | 56.43 | **26.88** | 82.80 | 52.00 |
| | Clippy | 52.86 | 29.10 | 56.48 | 53.76 | 31.80 | 73.07 | 55.72 | 26.03 | 82.60 | 51.27 |
| | **AdaGC** | **56.86** | **29.61** | 59.36 | **57.89** | 33.60 | **73.99** | **57.62** | 26.46 | **85.90** | **53.47** |
| Mixtral 8x1B | GlobalGC | 50.34 | 27.39 | **58.81** | 52.96 | **34.20** | 71.16 | 54.06 | **25.37** | 79.90 | 50.47 |
| | **AdaGC** | **53.83** | **28.42** | 58.69 | **55.66** | 33.80 | **73.07** | **54.14** | 25.12 | **81.80** | **51.61** |

### E.2    RESULTS OF OTHER BASELINE METHODS

Table 11: The Zero-Shot evaluation results of Llama-2 1.3B/7B models on standard benchmarks.

| Model | Method | ARC-E acc_norm | ARC-C acc_norm | BoolQ acc | HellaSw. acc_norm | OBQA acc_norm | PIQA acc_norm | W.G. acc | MMLU acc | SciQ acc_norm | Avg. |
|---|---|---|---|---|---|---|---|---|---|---|---|
| Llama-2 1.3B | WeSaR-GlobalGC | **43.56** | 25.17 | **59.94** | 45.08 | 30.00 | **70.29** | 52.96 | 22.90 | 65.80 | 46.19 |
| | SPAM | 42.05 | 24.83 | 59.60 | 42.82 | 30.00 | 69.31 | 52.17 | 23.02 | 66.40 | 45.58 |
| | ScaledEmbed-GlobalGC | 42.21 | **25.51** | 59.66 | 45.50 | **31.80** | 70.02 | **53.28** | 23.22 | 65.20 | 46.27 |
| | **AdaGC** | 42.09 | **25.51** | 58.01 | **47.29** | 30.40 | 69.70 | 52.33 | 22.98 | **68.70** | **46.33** |
| Llama-2 7B | WeSaR-GlobalGC | **49.75** | 27.22 | 56.12 | 55.38 | **33.80** | 73.39 | 56.27 | 23.02 | 71.40 | 49.59 |
| | SPAM | 48.53 | 25.77 | 60.34 | 51.89 | 32.60 | 72.03 | 54.54 | 22.95 | 71.00 | 48.85 |
| | ScaledEmbed-GlobalGC | 48.57 | 26.71 | **60.89** | 54.32 | 32.60 | 72.25 | 55.33 | **23.66** | 70.50 | 49.42 |
| | **AdaGC** | 49.58 | **28.92** | 57.28 | **57.94** | 32.80 | **74.32** | **58.09** | 23.62 | **76.60** | **51.01** |

In addition to the clipping-based baselines discussed in the main text, we also compare AdaGC with several recent methods that aim to improve the stability and generalization of large language model (LLM) training, including SPAM (Huang et al., 2025), Scaled Embed (Takase et al., 2023), and WeSaR (Nishida et al., 2024). The detailed results under the zero-shot setting and spike score are summarized in Table 11 and 12. The training dynamics are shown in Figures 8 and 9.

Table 12: Comparison of spike scores for various models under different methods.

| Model | Llama-2 1.3B | | | | Llama-2 7B | | | |
|---|---|---|---|---|---|---|---|---|
| Method | WeSaR-GlobalGC | SPAM | ScaledEmbed-GlobalGC | **AdaGC** | WeSaR-GlobalGC | SPAM | ScaledEmbed-GlobalGC | **AdaGC** |
| Total Steps | 9K | 9K | 9K | 9k | 9K | 9K | 9K | 9K |
| Num Spikes | 2 | 0 | 10 | 0 | 1 | 3 | 8 | 0 |
| Spike Score (%) | 0.0222 | **0.0000** | 0.1111 | **0.0000** | 0.0111 | 0.0333 | 0.0889 | **0.0000** |

Among these methods, SPAM is designed to stabilize training by adjusting the optimizer's behavior, while Scaled Embed and WeSaR focus on initialization or embedding scaling strategies to suppress loss spikes. Our experiments show that, although some of these methods can partly mitigate instability or improve certain metrics, AdaGC generally achieves higher stability and better final performance across model scales. Notably, while WeSaR is also effective at suppressing loss spikes, its reliance on special parameter initialization limits its applicability to from-scratch training. In contrast, AdaGC works reliably under both from-scratch and resumed training regimes, providing stronger flexibility. Overall, these results demonstrate AdaGC's superior robustness and generalization compared to other non-clipping baselines.

# F MORE VISUALIZATION RESULTS

## F.1 TRAINING DYNAMICS

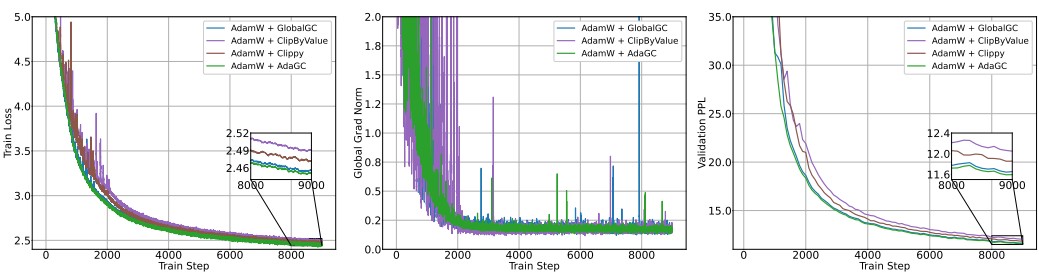

Figure 7: Llama-2 1.3B training dynamics of clipping methods.

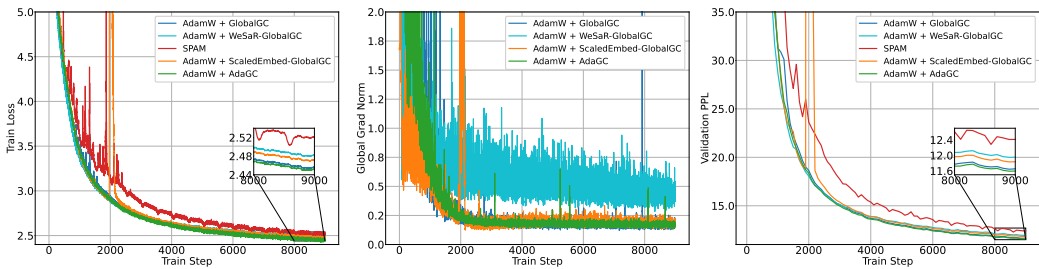

Figure 8: Llama-2 1.3B training dynamics of other baseline methods.

## F.2 OPTIMIZER STATE DYNAMICS

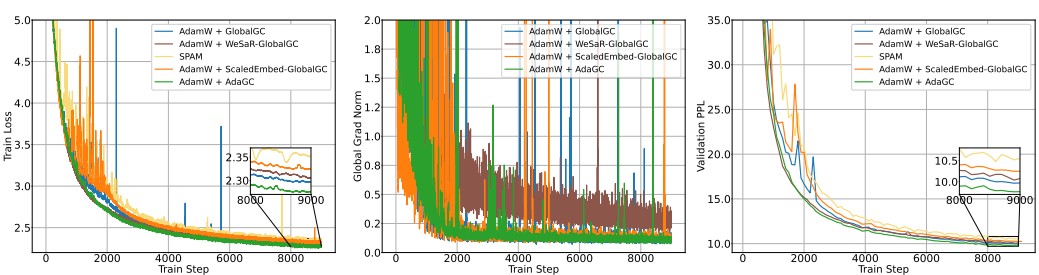

Figure 9: Llama-2 7B training dynamics of other baseline methods.

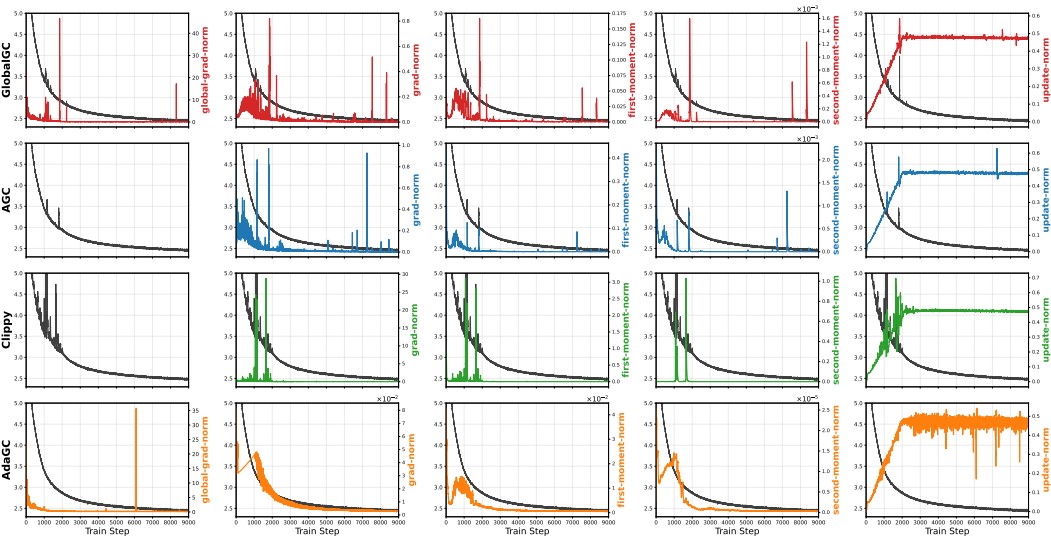

Figure 10: Visualization of the gradient norm, first-moment norm, second-moment norm, update norm, loss, and global gradient norm for the `embedding` of Llama-2 1.3B. Each row represents a different clipping method: the first row is GlobalGC, the second is AGC, the third is Clippy, and the fourth is our AdaGC. The black curve in each plot shows the loss trajectory.

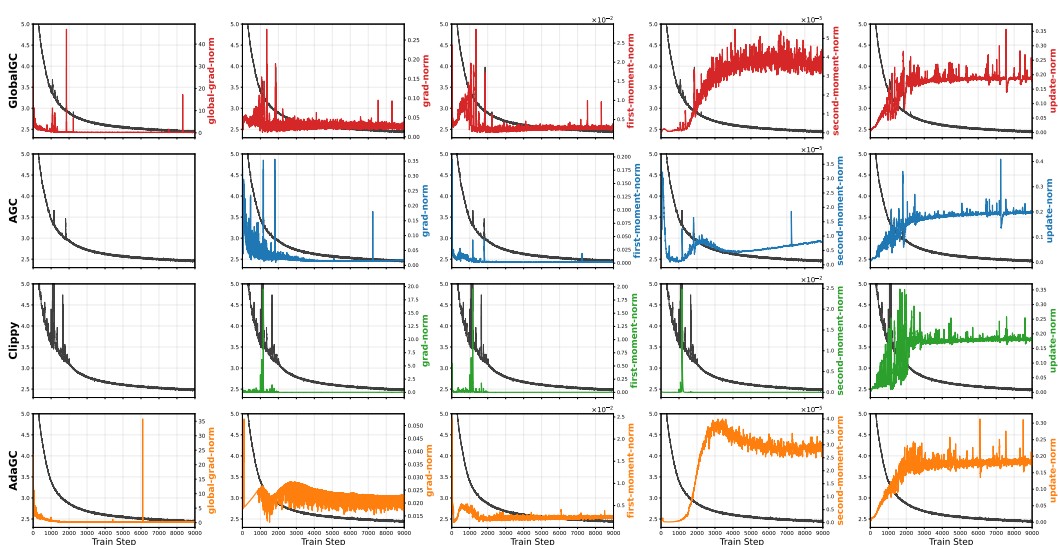

Figure 11: Visualization of the gradient norm, first-moment norm, second-moment norm, update norm, loss, and global gradient norm for the encoder_layers_3_self_attention_query_key_value of Llama-2 1.3B. Each row represents a different clipping method: the first row is GlobalGC, the second is AGC, the third is Clippy, and the fourth is our AdaGC. The black curve in each plot shows the loss trajectory.

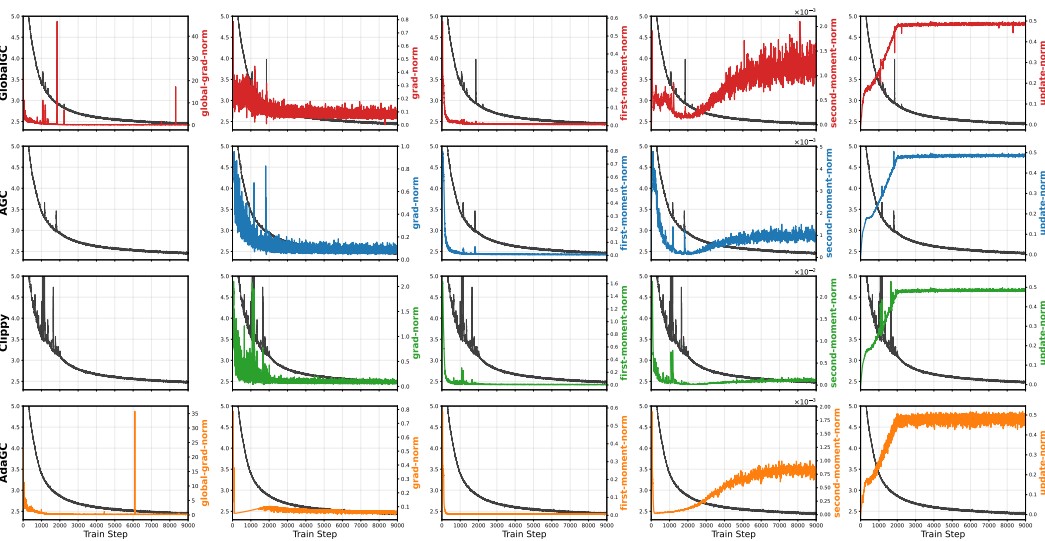

Figure 12: Visualization of the gradient norm, first-moment norm, second-moment norm, update norm, loss, and global gradient norm for the LMHead of Llama-2 1.3B. Each row represents a different clipping method: the first row is GlobalGC, the second is AGC, the third is Clippy, and the fourth is our AdaGC. The black curve in each plot shows the loss trajectory.

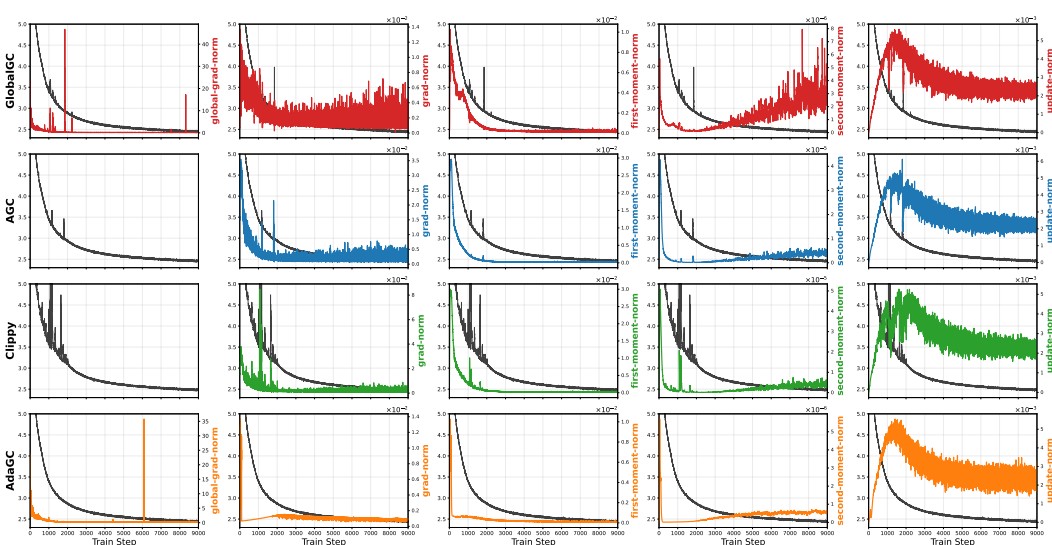

Figure 13: Visualization of the gradient norm, first-moment norm, second-moment norm, update norm, loss, and global gradient norm for the encoder_final_layernorm of Llama-2 1.3B. Each row represents a different clipping method: the first row is GlobalGC, the second is AGC, the third is Clippy, and the fourth is our AdaGC. The black curve in each plot shows the loss trajectory.

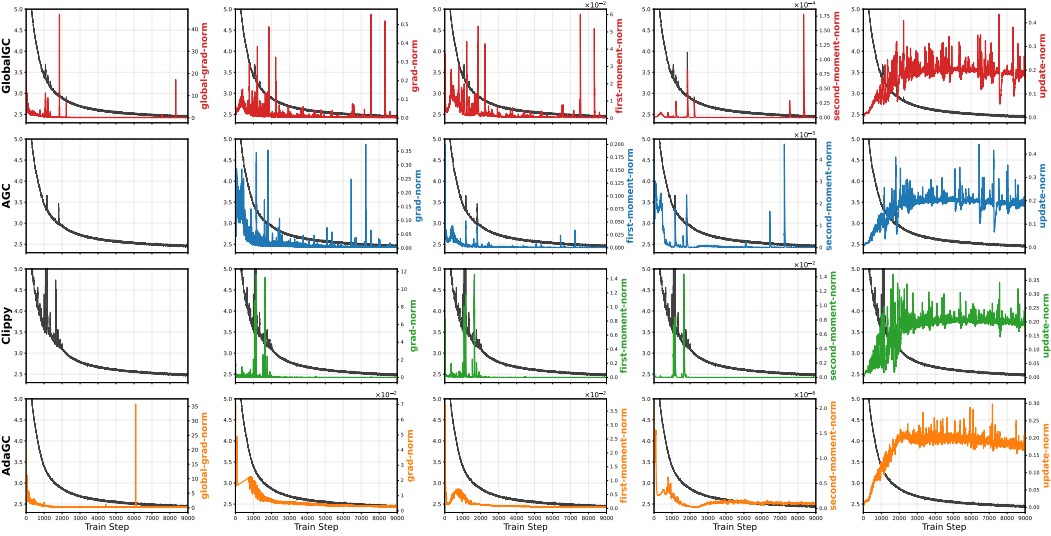

Figure 14: Visualization of the gradient norm, first-moment norm, second-moment norm, update norm, loss, and global gradient norm for the encoder_layers_0_self_attention_query_key_value of Llama-2 1.3B. Each row represents a different clipping method: the first row is GlobalGC, the second is AGC, the third is Clippy, and the fourth is our AdaGC. The black curve in each plot shows the loss trajectory.

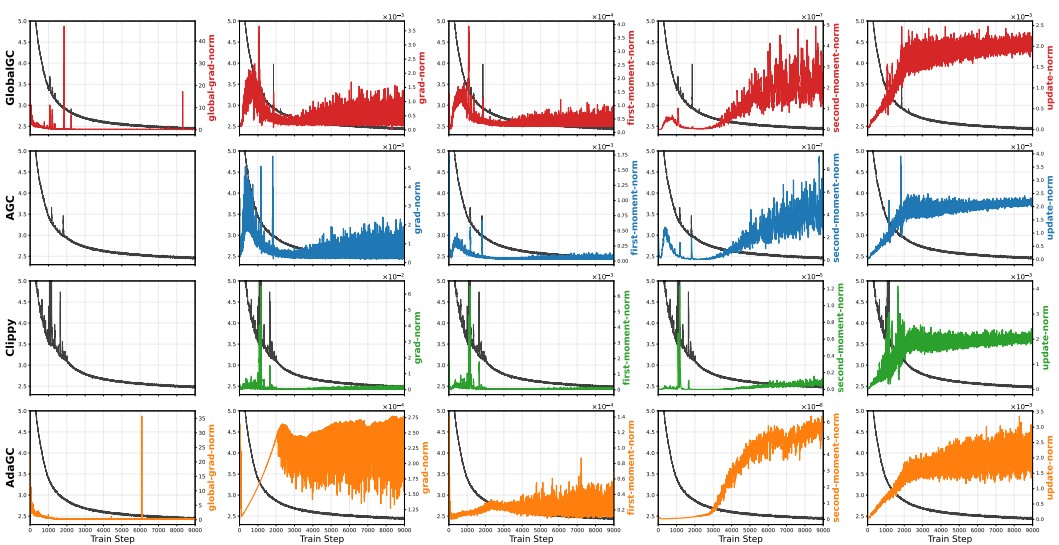

Figure 15: Visualization of the gradient norm, first-moment norm, second-moment norm, update norm, loss, and global gradient norm for the encoder_layers_23_input_layernorm of Llama-2 1.3B. Each row represents a different clipping method: the first row is GlobalGC, the second is AGC, the third is Clippy, and the fourth is our AdaGC. The black curve in each plot shows the loss trajectory.

