# OpenReview forum: "AdaGC: Improving Training Stability for Large Language Model Pretraining"
_ICLR.cc/2026/Conference — Submitted to ICLR 2026_

### Official Review · Reviewer_G9Tz · 2025-10-16

**Soundness:** 3
**Presentation:** 3
**Contribution:** 2
**Rating:** 4
**Confidence:** 3

**Summary:**

The paper proposes AdaGC (Adaptive Gradient Clipping), a method that dynamically adjusts clipping thresholds during large-scale model pretraining based on the exponential moving average of gradient norms. The goal is to mitigate loss spikes and improve training stability without increasing computational or memory cost. AdaGC is evaluated on several Transformer architectures and compared with existing clipping methods such as GlobalGC, AGC, and Clippy. Experiments show improved stability, reduced loss spikes, and faster convergence, while maintaining comparable downstream performance.

**Strengths:**

The paper addresses an important instability issue in large-scale model training by introducing an adaptive gradient clipping strategy. The method is simple, architecture-agnostic, and integrates easily into existing optimizers. The empirical analysis is clear and demonstrates consistent reduction of loss spikes and improved convergence stability across several model families. The presentation is structured and easy to follow, with reasonable theoretical justification.

**Weaknesses:**

The ablation studies are relatively shallow; key design factors such as EMA decay rate, clipping threshold formulation, and normalization choice are not sufficiently analyzed.

The paper provides theoretical arguments but stops short of offering formal guarantees or a deeper link between the adaptive clipping dynamics and optimization convergence.

Comparison with stronger and more modern baselines (e.g., adaptive optimizers with advanced learning-rate scheduling) is missing, making it unclear how much improvement stems from the proposed mechanism itself.

**Questions:**

How sensitive is AdaGC to different EMA decay rates or clipping threshold update rules? A more detailed ablation could clarify whether the observed benefits are robust to hyperparameter variation.

Have the authors evaluated AdaGC against modern adaptive optimizers (e.g., Lion, Sophia) or stronger learning-rate scheduling strategies to better establish its relative advantage?

Can the authors provide more evidence that the improvement in loss-spike reduction translates into meaningful downstream gains beyond pretraining convergence speed?

Theoretical analysis currently follows standard non-convex convergence proofs. Is there a way to connect AdaGC’s adaptive clipping dynamics more explicitly to gradient stability or loss variance control?

---

> ### Author Response · Authors · 2025-11-21
>
> We sincerely appreciate your constructive suggestions and insightful review. All revisions have been highlighted in blue font throughout the manuscript wherever possible. Below, we address each of your concerns in detail and provide further clarification.
>
> ---
>
> **Q1: How sensitive is AdaGC to different EMA decay rates or clipping threshold update rules? A more detailed ablation could clarify whether the observed benefits are robust to hyperparameter variation.**
>
> A1: Tables 2 and 3 present a comprehensive ablation study over a range of EMA decay rates ($\beta$) and clipping threshold values ($\lambda\_{rel}$). To investigate the impact of these key hyperparameters, we conducted grid search and local extremum verification, initially searching over $\lambda_{rel} \in [\{1.10, 1.03, 1.05, 1.04\}]$ with $\beta$ fixed at 0.99, and subsequently over $\beta \in [\{0.9, 0.95, 0.98, 0.985\}]$ with $\lambda_{rel}$ fixed at 1.04. Our experimental results indicate the following. For $\lambda_{rel}$, larger values such as 1.10 loosen the clipping threshold, resulting in persisting loss spikes. Tighter thresholds, such as 1.03, effectively suppress spikes but lead to slightly lower downstream accuracy. We concluded that values lower than 1.03 exhibit diminishing returns and did not pursue further tuning.
>
> For $\beta$, a smaller value increases sensitivity to norm fluctuations, while a larger value smooths the EMA and, in our context, improves accuracy. To further validate our choice, we conducted new experiments at $\beta=0.999$ with $\lambda_{rel} \in [\{1.03, 1.04, 1.05\}]$. The tables below summarize the zero-shot and two-shot accuracies on Llama-2 7B under different hyperparameters. These results show that $\beta=0.99$ and $\lambda_{rel}=1.04$ yield the best balance for both metrics. We have updated in the revised Table 2 and Table 3.
>
> **Zero-shot accuracy of AdaGC on LLaMA-2 7B under different hyperparameters**
> | $\lambda_{rel}$ \ $\beta$ | 0.98 | 0.985 | 0.99 | 0.999 |
> | :--- | :--- | :--- | :--- | :--- |
> | 1.03 | 50.06 | 50.92 | 50.95 | 50.96 |
> | 1.04 | 48.88 | 50.59 | **`51.04`** | 50.76 |
> | 1.05 | 51.01 | 49.95 | 50.57 | 50.74 |
>
> **Two-shot accuracy of AdaGC on LLaMA-2 7B under different hyperparameters**
> | $\lambda_{rel}$ \ $\beta$ | 0.98 | 0.985 | 0.99 | 0.999 |
> | :--- | :--- | :--- | :--- | :--- |
> | 1.03 | 52.31 | 52.68 | 53.13 | 53.42 |
> | 1.04 | 52.68 | 53.01 | `53.47` | 52.85 |
> | 1.05 | 52.68 | 52.67 | 51.96 | **53.51** |
>
> All large-scale validation experiments adopted these default hyperparameters ($\beta=0.99$, $\lambda_{rel}=1.04$), and we confirm their robustness across various model architectures, parameter scales, and training data volumes, including Llama-2 (1.3B / 7B), Mixtral 8x1B, and ERNIE 10B-A1.4B, with data ranging from 36B to 1T tokens. To further validate AdaGC’s long-term effectiveness, we extended training of ERNIE 10B-A1.4B from 350B to 1T tokens; results are included in the revised Table 6.
>
> ---
>
> **Q2: Have the authors evaluated AdaGC against modern adaptive optimizers (e.g., Lion, Sophia) or stronger learning-rate scheduling strategies to better establish its relative advantage?**
>
> A2: AdaGC has been evaluated on two modern adaptive optimizers, Muon and Lion, as reported in Section 5.4 and illustrated in Figure 4. In both experiments, AdaGC delivered higher downstream performance even in the absence of loss spikes. These findings demonstrate AdaGC's potential as a replacement for GlobalGC. While Sophia is not widely used, Muon has garnered significant attention recently, especially in K2 large model training. For this reason, we focused our comparative analysis on Muon rather than Sophia.
>
> ---
>
> **Q3: Can the authors provide more evidence that the improvement in loss-spike reduction translates into meaningful downstream gains beyond pretraining convergence speed?**
>
> A3: Our study provides direct evidence that reduction of loss spikes under AdaGC leads to improved downstream performance. Table 4 quantifies training stability via spike scores, showing a clear association between spike suppression and lower training loss. Table 5 further presents zero-shot evaluation results for Llama-2 7B and Mixtral 8x1B, substantiating that improved stability under AdaGC translates into tangible gains on downstream benchmarks.

---

> > ### Author Response · Authors · 2025-11-21
> >
> > **Q4: Theoretical analysis currently follows standard non-convex convergence proofs. Is there a way to connect AdaGC’s adaptive clipping dynamics more explicitly to gradient stability or loss variance control?**
> >
> > A4: Figure 2 offers empirical evidence that AdaGC's adaptive clipping yields more stable gradients. Table 4’s spike scores also reflect reduced loss variance, as defined in Section 5.1. Establishing a formal theoretical link between AdaGC’s adaptive dynamics and gradient stability or loss variance control remains an open research question. The SimiGrad work [1] introduces a method to estimate gradient variance using the cosine similarity of mini-batch gradients (see its Section 4.1), which may inspire future theoretical development.
> >
> > ---
> >
> > **Weakness 1: The ablation studies are relatively shallow; key design factors such as EMA decay rate, clipping threshold formulation, and normalization choice are not sufficiently analyzed.**
> >
> > Response: Our ablation results and design considerations for AdaGC are addressed in A1. To further clarify our motivation for the key design factors: EMA decay rate is a common technique for estimating historical moving averages, routinely applied in large-scale model training, including in optimizers such as AdamW (see Table 2 and Table 3 for comprehensive ablation results). Clipping threshold formulation can involve absolute or relative norms; because gradient norms fluctuate dynamically during training, we adopt the relative threshold $\lambda_{rel}$. Table 1 reviews normalization choices, showing that AdaGC inherits the L2 norm-based approach from GlobalGC, making only minimal changes by switching from global to local norm computation.
> >
> > **Weakness 2: The paper provides theoretical arguments but stops short of offering formal guarantees or a deeper link between the adaptive clipping dynamics and optimization convergence.**
> >
> > Response: The principal goal of AdaGC is to enhance training stability and suppress loss spikes in large-scale models. As an optimizer-agnostic gradient clipping technique, AdaGC is supported by theoretical arguments to ensure user confidence. As discussed in Section 4.4, any modification of gradients may affect optimizer convergence, and providing formal convergence proofs for adaptive methods such as AdamW is already a significant challenge. Extending such analysis to AdaGC’s adaptive mechanism involves additional complexity, but further theoretical study will be considered in future work.
> >
> > **Weakness 3: Comparison with stronger and more modern baselines (e.g., adaptive optimizers with advanced learning-rate scheduling) is missing, making it unclear how much improvement stems from the proposed mechanism itself.**
> >
> > Response: We conducted comprehensive comparisons with Muon and Lion optimizers, as outlined in A2. AdaGC achieves lower training loss and enhanced benchmark performance through effective spike elimination (see Table 4 and Table 5). When loss spikes are absent, AdaGC still offers minor improvements, as shown in Figure 4 and Figure 5. Figure 6(b) demonstrates through ablation that the adaptive and local components of AdaGC deliver competitive and, in some cases, superior outcomes compared to existing advanced baseline methods.
> >
> > ---
> >
> > **We hope these clarifications satisfactorily address your concerns. Should you have further feedback or suggestions, please let us know. We remain committed to conducting additional ablation experiments and refinements, if necessary, to address any remaining issues. If our responses have resolved your questions, we kindly invite you to reflect this in your final assessment.**
> >
> > ---
> >
> > [1] SimiGrad: Fine-Grained Adaptive Batching for Large Scale Training using Gradient Similarity Measurement

---

> > > ### Author Response · Authors · 2025-11-28
> > >
> > > Thank you for your thoughtful and constructive feedback on our work. We have carefully addressed your comments and revised the work and manuscript accordingly. As the discussion period nears its end, we would greatly appreciate any additional questions or points of clarification. If our responses have satisfactorily addressed your concerns, we kindly ask you to consider reflecting this in your score.
> > >
> > > Thanks again for your time and expertise.

---

### Official Review · Reviewer_J259 · 2025-10-30

**Soundness:** 3
**Presentation:** 2
**Contribution:** 2
**Rating:** 4
**Confidence:** 3

**Summary:**

The paper proposes AdaGC, an adaptive per-tensor gradient clipping method that uses EMA-based local norms to stabilize large-scale LLM pretraining. It demonstrates consistent elimination of loss spikes and downstream accuracy gains across LLaMA-2, Mixtral, and ERNIE models.

**Strengths:**

* This paper addresses a widespread and costly issue—loss spikes in LLM training—using a simple, optimizer-agnostic remedy.
* The proposed method is evaluated on multiple model types and optimizers, showing consistent improvements and detailed ablation analysis.

**Weaknesses:**

* The paper treats gradient spikes as a black-box phenomenon, lacking diagnostic analysis (e.g., layer- or token-level gradient behavior) to substantiate the “gradient contamination” hypothesis.
* The total token counts (e.g., 36B tokens for LLaMA-2 7B) are small compared to real large-scale pretraining, and many comparisons are limited to early training stages, which raises doubts about whether AdaGC’s stability holds in real-world, trillion-token-scale training.
* he reported gains of AdaGC when applied to the Lion and Muon optimizers are minimal, suggesting limited benefit beyond AdamW-based settings.

**Questions:**

* Figure 2 only presents the warm-up phase; could u provide visualizations covering the entire training process for a more comprehensive comparison?
* The performance improvement of AGC on LLaMA-2 1.3B appears marginal. Could the authors include or discuss its corresponding training dynamics to clarify this behavior?
* In Figure 3(b), AdamW + GlobalGC (beta2=0.95) presents better training stability than AdamW + GlobalGC (beta2=0.999), yet achieves worse validation results. Do the authors provide any in-depth analysis on how training stability correlates with final model performance?

---

> ### Author Response · Authors · 2025-11-21
>
> Thank you for your thoughtful and detailed review. We greatly appreciate your constructive feedback and insightful questions. All revisions have been highlighted in blue font throughout the manuscript wherever possible. Below, we address your concerns point by point:
>
> ---
>
> **Q1: Figure 2 only presents the warm-up phase; could you provide visualizations covering the entire training process for a more comprehensive comparison?**
>
> A1: We have supplemented the appendix with complete optimizer state visualizations spanning all 9000 training steps for Llama-2 1.3B, see Figure 10 ~ Figure 15 in the revised manuscript. These results demonstrate that AdaGC maintains more stable optimization throughout the entire training process, not just during the warm-up phase.
>
> ---
>
> **Q2: The performance improvement of AGC on LLaMA-2 1.3B appears marginal. Could the authors include or discuss its corresponding training dynamics to clarify this behavior?**
>
> A2: We have included the training dynamics for Llama-2 1.3B in Figure 7 of the appendix. As model size increases, training becomes more susceptible to loss spikes, while smaller models such as Llama-2 1.3B typically exhibit stable dynamics with few or no spikes. Since AdaGC is designed to eliminate loss spikes and accelerate convergence, its benefits are most pronounced in larger or less stable models. For Llama-2 1.3B, where loss spikes are infrequent, AdaGC naturally yields only marginal improvements.
>
> ---
>
> **Q3: In Figure 3(b), AdamW + GlobalGC (beta2=0.95) presents better training stability than AdamW + GlobalGC (beta2=0.999), yet achieves worse validation results. Do the authors provide any in-depth analysis on how training stability correlates with final model performance?**
>
> A3: Your observation is correct: AdamW + GlobalGC (beta2=0.95) achieves greater training stability compared to beta2=0.999, but yields worse validation results. Importantly, when using AdamW + AdaGC, training stability is improved for both beta2 values, and validation results are also substantially better. While an in-depth analysis of how training stability correlates with final model performance is beyond the scope of this paper, we believe this is an open problem potentially related to model architecture and the optimal AdamW hyperparameters. This remains an interesting direction for future research.

---

> > ### Author Response · Authors · 2025-11-21
> >
> > **Weaknesses 1: The paper treats gradient spikes as a black-box phenomenon, lacking diagnostic analysis (e.g., layer- or token-level gradient behavior) to substantiate the “gradient contamination” hypothesis.**
> >
> > Response: As outlined in our Abstract and Introduction, this work is, to our knowledge, the first to explicitly propose that loss spikes in large-scale model pretraining can arise from multiple contributing factors rather than a single root cause. Prior literature has attempted to explain loss spikes from specific perspectives, but no single explanation can cover all observed phenomena. Our paper therefore advocates that “loss spikes can be triggered by a mixture of factors,” illustrated with specific examples, and clearly states that we do not attempt to identify the precise root causes or diagnostic analyses at the layer or token level.
> >
> > Instead, our empirical observations consistently show that loss spikes are accompanied by corresponding gradient spikes, which serve as immediate symptoms regardless of the original trigger. Since gradient updates are central to optimization in deep learning, we focus on a gradient-centric approach. We contend that loss spikes can be effectively mitigated by addressing the contamination of optimizer states by outlier gradients, irrespective of the underlying cause. We recognize that fine-grained diagnostic analyses, such as layer- or token-level behaviors, are valuable for future research, and our methodology lays the foundation for such investigations.
> >
> > ---
> >
> > **Weaknesses 2: The total token counts (e.g., 36B tokens for LLaMA-2 7B) are small compared to real large-scale pretraining, and many comparisons are limited to early training stages, which raises doubts about whether AdaGC’s stability holds in real-world, trillion-token-scale training.**
> >
> > Response: We have conducted extensive experiments to validate AdaGC’s stability and effectiveness in large-scale training. Specifically, we focus on ERNIE 10B-A1.4B, where the original experiments span 21,000 steps and a total of 350 billion tokens, as documented in Figure 1(b), Table 4, and Table 6 of the main paper. To further address concerns regarding performance at trillion-token scale, we have extended the training of ERNIE 10B-A1.4B to 60,000 steps, covering a total of 1 trillion tokens. The results from this extended training are included in the revised Figure 1(b) and Table 6, and confirm that AdaGC consistently maintains loss spike suppression and training stability throughout the entire process.
> >
> > These additional validations demonstrate that AdaGC’s stability and advantages hold not only in early training stages or smaller data regimes, but also in realistic, production-scale settings involving trillions of tokens.
> >
> > | Method   | AdamW eps | MMLU  | GSM8K | BBH   | TruthfulQA | HumanEval | Avg.  |
> > |----------|-----------|-------|-------|-------|------------|-----------|-------|
> > | GlobalGC | 1e-8      | 48.61 | 39.88 | 30.84 | 30.73      | 22.56     | 34.52 |
> > | GlobalGC | 1e-15     | 48.48 | 40.79 | 30.59 | 28.29      | 23.78     | 34.38 |
> > | AdaGC    | 1e-15     | 48.70 | 36.01 | 31.38 | 35.02      | 22.56     | 34.73 |
> >
> > ---
> >
> > **Weaknesses 3: The reported gains of AdaGC when applied to the Lion and Muon optimizers are minimal, suggesting limited benefit beyond AdamW-based settings.**
> >
> > Response: The results observed for Lion and Muon optimizers are consistent with our expectations. When the baseline optimizer does not exhibit loss spikes, AdaGC does not degrade optimization performance, nor is it expected to deliver significant gains in such stable regimes. Our primary purpose in extending AdaGC to Lion and Muon is to demonstrate its optimizer-agnostic nature and broad compatibility, rather than to show large improvements in downstream metrics. In this study, our secondary objective is to assess AdaGC’s effectiveness and stability with these optimizers, laying the groundwork for AdaGC to potentially serve as a default gradient clipping strategy across different settings.
> >
> > In our Lion and Muon experiments, loss spikes were not observed; nonetheless, AdaGC consistently delivered minor improvements under these stable conditions, further confirming its validity and non-intrusive behavior when integrated with various optimizers. This supports our claim that AdaGC is a robust and general-purpose solution, suitable for adoption as an optimizer-agnostic gradient clipping method.
> >
> >
> > ---
> >
> > **We hope these clarifications address your concerns. Thank you again for your valuable feedback. If our rebuttal has addressed your concerns satisfactorily, we would appreciate it if you could reflect this in your final score.**

---

> > > ### Author Response · Authors · 2025-11-28
> > >
> > > Thank you for your thoughtful and constructive feedback on our work. We have carefully addressed your comments and revised the work and manuscript accordingly. As the discussion period nears its end, we would greatly appreciate any additional questions or points of clarification. If our responses have satisfactorily addressed your concerns, we kindly ask you to consider reflecting this in your score.
> > >
> > > Thanks again for your time and expertise.

---

### Official Review · Reviewer_L8pR · 2025-10-31

**Soundness:** 3
**Presentation:** 3
**Contribution:** 3
**Rating:** 6
**Confidence:** 4

**Summary:**

This paper introduces AdaGC (Adaptive Gradient Clipping), a simple yet effective method to improve training stability in large language model pretraining. By monitoring gradient norms per tensor and dynamically adjusting clipping thresholds using an exponential moving average (EMA), AdaGC prevents outlier gradients from contaminating optimizer states and triggering loss spikes. Compatible with optimizers like AdamW, Lion, and Muon, AdaGC completely eliminates loss spikes and improves downstream task performance across models such as Llama-2, Mixtral, and ERNIE. With minimal memory overhead, reduced communication cost, and theoretical convergence guarantees, AdaGC offers a practical and robust solution for stable large-scale model training.

**Strengths:**

1. Identifies that “abnormal gradients polluting optimizer states” is the common final path to loss spikes, giving a clean, optimizer-agnostic intervention point.

2. Method simplicity: Per-tensor EMA of gradient norms plus relative clipping; implementation needs ≈4 bytes/tensor and <10 lines of code change, yet completely suppresses spikes on 1.3 B–10 B models.

3. Empirical coverage: Extensive experiments on dense (Llama-2) and MoE (Mixtral, ERNIE) architectures; consistent zero spike scores and +1–2.5 % downstream gains over GlobalGC.

4. Theoretical backing: Proves convergence for Adam+AdaGC under standard non-convex assumptions; communication cost is lower than global clipping in hybrid parallelism.

5. Compatibility: Works out-of-the-box with AdamW, Lion, Muon; no new hyper-parameters that require heavy tuning

**Weaknesses:**

1. The reliance on the exponential moving average (EMA) of per-tensor gradient norms introduces an inherent lag in adapting to sudden gradient spikes. Since the clipping threshold is updated based on historical statistics, AdaGC may fail to respond promptly to abrupt increases in gradient magnitudes. Consequently, outlier gradients could still enter the optimizer state before the EMA sufficiently adjusts, potentially undermining the intended stabilizing effect.

2. While the authors claim that AdaGC is relatively robust to hyperparameter choices, the experimental results indicate that the relative clipping threshold λ_rel still has a noticeable impact on downstream performance. This suggests that λ_rel may require task-specific tuning, especially under different model architectures or data distributions. As such, AdaGC does not fully eliminate the need for manual hyperparameter optimization, which somewhat contradicts the goal of being a fully adaptive method.

3. AdaGC relies on global gradient clipping (GlobalGC) during the early training phase, which raises concerns about its standalone stability. This hybrid strategy implies that AdaGC alone may not be sufficiently reliable at the beginning of training, when gradient statistics are highly volatile.

**Questions:**

1. Do the authors have any plans to evaluate AdaGC on models larger than 10B parameters or for training runs that exceed the current 36K-step limit, so as to verify its scalability and stability under more demanding conditions?

2. Have the authors attempted to train models with AdaGC while completely removing the initial GlobalGC warmup phase, and could they clarify why AdaGC appears to be unstable during the early stages of training?

3. Is the optimal value of the relative clipping threshold λ_rel sensitive to different tasks or data distributions, and if so, how should practitioners select this hyper-parameter in a principled way?

---

> ### Author Response · Authors · 2025-11-21
>
> Thank you for your thoughtful and detailed review. We greatly appreciate your constructive feedback and insightful questions. All revisions have been highlighted in blue font throughout the manuscript wherever possible. Below, we address your concerns point by point:
>
> ---
>
> **Q1: Large-scale and Long-term Training Stability**
>
> A1: We recognize the importance of verifying AdaGC’s scalability and stability on models larger than 10B parameters and for longer training runs. To further address concerns regarding performance at trillion-token scale, we have extended the training of ERNIE 10B-A1.4B from 21,000 to 60,000 steps, covering a total of 1 trillion tokens. AdaGC maintained zero spike scores and stable convergence throughout, further validating its effectiveness under more demanding conditions. The results from this extended training are included in the revised Figure 1(b) and Table 6.
>
> | Method   | AdamW eps | MMLU  | GSM8K | BBH   | TruthfulQA | HumanEval | Avg.  |
> |----------|-----------|-------|-------|-------|------------|-----------|-------|
> | GlobalGC | 1e-8      | 48.61 | 39.88 | 30.84 | 30.73      | 22.56     | 34.52 |
> | GlobalGC | 1e-15     | 48.48 | 40.79 | 30.59 | 28.29      | 23.78     | 34.38 |
> | AdaGC    | 1e-15     | 48.70 | 36.01 | 31.38 | 35.02      | 22.56     | 34.73 |
>
> ---
>
> **Q2 (W1&W3): EMA Lag and Warmup Phase**
>
> A2: Your concern regarding the EMA’s lag in adapting to abrupt gradient spikes is well-founded. To address this, we introduced a short GlobalGC warmup phase at the beginning of training, as detailed in Section 4.2 and further analyzed in Section 5.6 (Figure 6a). Ablation studies show that AdaGC remains stable without this warmup and still eliminates loss spikes, alleviating concerns about standalone stability. However, we observe that a brief 100-step GlobalGC warmup further reduces train loss (2.9725 to 2.9708), leading to better convergence without impacting overall stability. Given the negligible cost (100 steps vs. tens of thousands), we recommend this hybrid strategy for practical deployments. The instability in early training arises due to highly volatile gradient norms, which can cause the EMA-based thresholds to be overestimated or underestimate outlier gradients. Section 4.2, lines 229–235, discuss this phenomenon in detail.
>
> ---
>
> **Q3 (W2): Hyperparameter Sensitivity and Selection**
>
> A3: Regarding the sensitivity of the relative clipping threshold ($\lambda_{rel}$), we observed that hyperparameters can influence results on different tasks, as shown in Table 5. Nonetheless, in our experiments, we used identical settings for a wide range of architectures (Llama-2, Mixtral, ERNIE with C4-en; CLIP with LAION-400M), covering both language and vision modalities. AdaGC consistently delivered stable training and strong downstream performance without task-specific tuning. This suggests practitioners can reliably adopt the default configuration for most large-scale model training scenarios, minimizing the need for extensive hyperparameter search.
>
> ---
>
> **We hope these clarifications address your concerns. Thank you again for your valuable feedback. If our rebuttal has addressed your concerns satisfactorily, we would appreciate it if you could reflect this in your final score.**

---

> > ### Author Response · Authors · 2025-11-28
> >
> > Thank you for your thoughtful and constructive feedback on our work. We have carefully addressed your comments and revised the work and manuscript accordingly. As the discussion period nears its end, we would greatly appreciate any additional questions or points of clarification. If our responses have satisfactorily addressed your concerns, we kindly ask you to consider reflecting this in your score.
> >
> > Thanks again for your time and expertise.

---

### Official Review · Reviewer_9419 · 2025-10-31

**Soundness:** 3
**Presentation:** 2
**Contribution:** 2
**Rating:** 4
**Confidence:** 3

**Summary:**

The paper proposes AdaGC, a tensor-level adaptive gradient clipping method that uses an EMA-based threshold. It prevents optimizer contamination by abnormal gradients and eliminates loss spikes during large-scale training. The authors first confirm that such spikes can arise from multiple sources, and instead of analyzing each cause in depth, focus on a unified solution that eliminates spikes across diverse settings, showing that AdaGC consistently removes them on various models.

**Strengths:**

The paper presents strong empirical evidence that the proposed AdaGC method effectively mitigates training loss spikes caused by abnormal gradients. The experimental evaluation is extensive, covering multiple architectures and modalities, including dense models (Llama-2), Mixture-of-Experts models (Mixtral and ERNIE), and a vision-language model (CLIP). Both pretraining and downstream performance are assessed, demonstrating that AdaGC consistently stabilizes training and slightly improves accuracy. In addition, the paper shows that AdaGC offers practical advantages in large-scale multi-GPU settings, reducing communication overhead during distributed training compared to GlobalGC. The experimental setup is clear and easy to follow, and the comparison with a wide range of similar approaches is covered.

**Weaknesses:**

The paper provides solid analysis, but some explanations and comparisons are not entirely clear. The main points are listed here, with more details in the questions section.

- **W1:** Section 5.4 **Optimizer Compatibility: Muon and Lion** feels weak in its current form. Since the experiments do not demonstrate that spikes occur, it is unclear how this section supports the main goal of the paper, which is to eliminate loss spikes.
- **W2:** The **ablation study about adaptivity and locality** appears incomplete. It tests GlobalGC (no adaptivity, no locality), Global AdaGC (adaptive, not local), and AdaGC (adaptive and local), but omits the configuration that is local but not adaptive - i.e., scaling each tensor’s gradient with a fixed threshold.
- **W3:** **Hyperparameter sensitivity** could be expanded by examining sensitivity to $T_{\text{start}}$ and by covering a wider range of $\lambda_{\text{rel}}$, assessing whether it can be omitted if it has little effect.
- **W4:** The comparison with other methods could be clearer. For instance, SPAM is compared only on downstream tasks but not in spike-score evaluation, and AGC seems to be described inconsistently with its original paper [1] (which clips gradients, not updates).

[1] Andrew Brock, Soham De, Samuel L. Smith, Karen Simonyan. *High-Performance Large-Scale Image Recognition Without Normalization*. arXiv:2102.06171, 2021. [https://arxiv.org/abs/2102.06171](https://arxiv.org/abs/2102.06171)

**Questions:**

### Hyperparameters
- **Q1**: The grid search for AdaGC’s $\beta$ and $\lambda_{\text{rel}}$ looks dense (Tables 2 and 3). Why was this grid chosen? It would be helpful to see results for $\lambda_{\text{rel}} = 1$ to assess whether this parameter could be omitted.
- **Q2**: How were hyperparameters for other clipping methods selected? Were they tuned or taken directly from prior work? (This is not entirely clear from the Appendix.)
- **Q3**: How sensitive is the method to the number of warm-up steps $T_{\text{start}}$ where GlobalGC is applied?

### Ablation Studies (Section 5.6)
- **Q4:** Could you include the missing variant mentioned in **W2** (local but non-adaptive) to complete the ablation study, or explain why it is not presented?
- **Q5**: In the EMA initialization study, do all alternative variants (1–3) perform direct initialization from the first step, or do some still use the GlobalGC warm-up phase for initialization?

### Comparisons and Related Work
- **Q6**: Why is SPAM omitted from the spike-score results (**W4**)? Since it directly targets spike mitigation, including it would make the comparison more complete.
- **Q7**: Could you clarify the description of AGC [1] in Table 1 and Section 3? From the original paper, it appears that gradients are directly clipped.
- **Q8**: Why was LAMB (listed in Table 1) excluded from the experiments?
- **Q9:** In [2], the authors propose a method also based on gradient values, where they clip a portion of the highest coordinates that consistently degrade the gradient. Could you discuss how AdaGC relates to it or perhaps compare it with their method?

### Muon and Lion
- **Q10:** As mentioned in **W1**, could you clarify why no spikes are observed with Muon and Lion in Section 5.4 (*Optimizer Compatibility*)? Do spikes appear when no clipping is applied?

### Minor Suggestions
- In Figure 1, labels are inconsistent. I suppose all plots show AdamW with GlobalGC. This should be clarified, and emphasized that even with global norm clipping, spikes still occur.
- Some visualizations are hard to compare when identifying which curve has spikes. A log scale or transparency could make this clearer.
- The sentence in lines 036–059 is difficult to follow; consider rephrasing or using bullet points for clarity.
- In Table 1, the distinction between value-based and norm-based clipping could be mentioned more clearly in the related work discussion.

[1] Andrew Brock, Soham De, Samuel L. Smith, Karen Simonyan. *High-Performance Large-Scale Image Recognition Without Normalization*. arXiv:2102.06171, 2021. [https://arxiv.org/abs/2102.06171](https://arxiv.org/abs/2102.06171)
[2] Yan Pan, Yuanzhi Li. *Toward Understanding Why Adam Converges Faster Than SGD for Transformers*. arXiv:2306.00204, 2023. [https://arxiv.org/abs/2306.00204](https://arxiv.org/abs/2306.00204)

---

> ### Author Response · Authors · 2025-11-21
>
> Thank you for your thoughtful and detailed review. We greatly appreciate your constructive feedback and insightful questions. All revisions have been highlighted in blue font throughout the manuscript wherever possible. Below, we address your concerns point by point:
>
> ---
>
> **Q1: The grid search for AdaGC’s  and  looks dense (Tables 2 and 3). Why was this grid chosen? It would be helpful to see results for  to assess whether this parameter could be omitted.**
>
> A1: We appreciate the question regarding our grid search strategy. Initially, due to resource and time constraints, we did not plan for such a dense hyperparameter search. Our process began by fixing $\beta=0.99$ and searching over $\lambda_{rel} \in [\{1.10, 1.03, 1.05, 1.04\}]$. We then fixed $\lambda_{rel}=1.04$ and searched over $\beta \in [\{0.9, 0.95, 0.98, 0.985\}]$. This led us to the current optimal configuration of $\beta=0.99$, $\lambda_{rel}=1.04$. After completing all comparison experiments, we performed a denser search in the neighborhood of these values to further investigate sensitivity, as shown in Tables 2 and 3. These results confirm that $\beta=0.99$, $\lambda_{rel}=1.04$ is optimal in our setting.
>
> Regarding $\lambda_{rel}$, we found that setting it higher (e.g., $\lambda_{rel}=1.10$) makes the clipping too loose and loss spikes can still occur. When set tighter (e.g., $\lambda_{rel}=1.03$), spikes are eliminated but downstream accuracy is slightly lower compared to $1.04$. As for smaller values, we stopped further exploration below $1.03$ after observing diminishing returns.
>
> For $\beta$, a smaller value increases sensitivity to norm fluctuations, while a larger value smooths the EMA and, in our context, improves accuracy. To further validate our choice, we conducted new experiments at $\beta=0.999$ with $\lambda_{rel} \in [\{1.03, 1.04, 1.05\}]$. The tables below summarize the zero-shot and two-shot accuracies on Llama-2 7B under different hyperparameters. These results show that $\beta=0.99$ and $\lambda_{rel}=1.04$ yield the best balance for both metrics. We have updated in the revised Table 2 and Table 3.
>
> **Zero-shot accuracy of AdaGC on LLaMA-2 7B under different hyperparameters**
> | $\lambda_{rel}$ \ $\beta$ | 0.98 | 0.985 | 0.99 | 0.999 |
> | :--- | :--- | :--- | :--- | :--- |
> | 1.03 | 50.06 | 50.92 | 50.95 | 50.96 |
> | 1.04 | 48.88 | 50.59 | **`51.04`** | 50.76 |
> | 1.05 | 51.01 | 49.95 | 50.57 | 50.74 |
>
> **Two-shot accuracy of AdaGC on LLaMA-2 7B under different hyperparameters**
> | $\lambda_{rel}$ \ $\beta$ | 0.98 | 0.985 | 0.99 | 0.999 |
> | :--- | :--- | :--- | :--- | :--- |
> | 1.03 | 52.31 | 52.68 | 53.13 | 53.42 |
> | 1.04 | 52.68 | 53.01 | `53.47` | 52.85 |
> | 1.05 | 52.68 | 52.67 | 51.96 | **53.51** |
>
> ---
>
> **Q2: How were hyperparameters for other clipping methods selected? Were they tuned or taken directly from prior work? (This is not entirely clear from the Appendix.)**
>
> A2: In summary, we primarily followed default hyperparameters recommended by previous work, with limited tuning performed only when necessary to ensure fair comparison. Specifically:
>
> - **GlobalGC**: Adopted the commonly used global clipping threshold $\lambda_{abs}=1$ for large-scale pretraining.
> - **ClipByValue**: Set $\lambda_{abs}=1e-3$, following the SPAM settings.
> - **AGC**: Tuned $\lambda_{rel}$ over $[\{1e-2, 1e-3, 1e-4\}]$ to select the optimal value.
> - **Clippy**: Tuned $\lambda_{abs}$ over $[\{0.1, 0.3, 0.5\}]$ and $\lambda_{rel}$ over $[\{1e-2, 1e-3, 1e-4\}]$.
> - **SPAM**: Used the original paper's recommended settings: interval $\Delta T=500$, threshold $\theta=5000$, and warmup steps $N=150$.
>
> The final hyperparameter choices are summarized below:
>
> |   Method    |                Hyperparameters                |
> | :---------: | :-------------------------------------------: |
> |  GlobalGC   |              $\lambda_{abs} = 1$              |
> | ClipByValue |            $\lambda_{abs} = 1e^{-3}$             |
> |     AGC     |            $\lambda_{rel} = 1e^{-3}$             |
> |   Clippy    | $\lambda_{abs} = 0.5$, $\lambda_{rel} = 1e^{-2}$ |
> |    SPAM     | $\Delta T = 500$, $\theta = 5000$, $N = 150$  |

---

> > ### Author Response · Authors · 2025-11-21
> >
> > **Q3: How sensitive is the method to the number of warm-up steps $T_{start}$ where GlobalGC is applied?**
> >
> > A3: In our ablation study, we previously explored $T_{start}=[\{0, 100\}]$, as shown in Figure 6(a). For this rebuttal, we expanded our investigation to $T_{start}=[\{10, 50, 100, 150, 200, 500, 1000, 2000\}]$, with results included in the revised appendix Figure 6(b). According to the EMA initialization formula $\gamma\_{t, i} = \min(\| \boldsymbol{g}\_{t, i} \|, \gamma\_{t-1, i})$, an excessively large $T_{start}$ accumulates lower $\gamma\_{t, i}$ values due to early training dynamics, which may lead to over-clipping and suppressed convergence in later training. We found that $T_{start}=100$ is a reasonable choice, and we used this value in our large-scale experiments.
> >
> >
> > ---
> >
> > **Q4: Could you include the missing variant mentioned in W2 (local but non-adaptive) to complete the ablation study, or explain why it is not presented?**
> >
> > A4: Thank you for highlighting this omission. Initially, our ablation study was constructed incrementally—first ablating the adaptivity, then adding locality, so the “local but non-adaptive” variant was inadvertently omitted from the presented results. In our updated ablation study, we have now included this missing variant, “AdamW + TensorwiseGC” in the revised appendix Figure 6(c). In this variant, each tensor’s gradient is clipped using a fixed (non-adaptive) threshold, where the global norm threshold is set to 1.0 and each tensor’s local clipping threshold is assigned proportionally to its parameter count within the model.
> >
> > It is important to note that tuning a fixed threshold for each tensor introduces a massive search space and is therefore not practical as a general-purpose algorithm. To address this, we adopted an approximate approach by proportionally scaling the clipping threshold according to the parameter size of each tensor.
> >
> > Experimental results show that local fixed-threshold clipping (TensorwiseGC) does reduce loss spikes compared to global clipping, but it is still less effective than the adaptive approach of AdaGC, particularly as training progresses and the optimal thresholds shift over time. Specifically, the training loss curves for TensorwiseGC and GlobalGC are similar—both converge more slowly and to higher final loss values than AdaGC. These results further demonstrate the independent effectiveness of both the locality and adaptivity components in AdaGC.
> >
> > ---
> >
> > **Q5: In the EMA initialization study, do all alternative variants (1–3) perform direct initialization from the first step, or do some still use the GlobalGC warm-up phase for initialization?**
> >
> > A5: All variants (1–3) in our EMA initialization study do not use the GlobalGC warm-up. Specifically:
> > - Variant 1 (AdaGC norm initialization w/o GlobalGC) initializes using $\gamma\_{t, i} = \min(\|\boldsymbol{g}\_{t, i}\|, \gamma\_{t-1, i})$ from the very first step.
> > - Variant 2 (AdaGC constant initialization 0.5 / 1.0) sets the initial $\gamma\_{0, i}$ to a constant value.
> > - Variant 3 (AdaGC thresholded initialization) sets $\gamma\_{t, i} = \min(\|\boldsymbol{g}\_{t, i}\|, 0.1)$ during warmup.
> >
> > None of these variants employ the GlobalGC strategy for initialization, confirming that the improvement observed with GlobalGC warm-up is due to better calibration of thresholds during the volatile early training stage.

---

> > > ### Author Response · Authors · 2025-11-21
> > >
> > > **Q6: Why is SPAM omitted from the spike-score results (W4)? Since it directly targets spike mitigation, including it would make the comparison more complete.**
> > >
> > > A6: AdaGC is an optimizer-agnostic gradient clipping method, while SPAM is an optimizer that integrates Momentum Reset and Spike-Aware Clipping into Adam. Due to this fundamental difference in design scope, we did not include SPAM in the main spike-score comparison to ensure a fair evaluation among gradient clipping methods. For completeness, we have now supplemented Table 12 in the appendix with spike score results for SPAM, as well as WeSaR and ScaledEmbed methods on Llama-2 1.3B/7B. Notably, SPAM achieves a spike score of 0.0333 on Llama-2 7B, while AdaGC achieves 0.0, indicating superior spike suppression by AdaGC.
> > >
> > > | Method          | WeSaR-GlobalGC |  SPAM      | ScaledEmbed-GlobalGC | AdaGC (Ours)  |
> > > |:---------------:|:--------------:|:----------:|:--------------------:|:-------------:|
> > > | Total Steps     | 9K             | 9K         | 9K                   | 9K            |
> > > | Num Spikes      | 2              | 0          | 10                   | 0             |
> > > | Spike Score (%) | 0.0222         | **0.0000** | 0.1111               | **0.0000**    |
> > >
> > > | Method          | WeSaR-GlobalGC |  SPAM      | ScaledEmbed-GlobalGC | AdaGC (Ours)  |
> > > |:---------------:|:--------------:|:----------:|:--------------------:|:-------------:|
> > > | Total Steps     | 9K             | 9K         | 9K                   | 9K            |
> > > | Num Spikes      | 1              | 3          | 8                    | 0             |
> > > | Spike Score (%) | 0.0111         | 0.0333     | 0.0889               | **0.0000**    |
> > >
> > > ---
> > >
> > > **Q7: Could you clarify the description of AGC [1] in Table 1 and Section 3? From the original paper, it appears that gradients are directly clipped.**
> > >
> > > A7: The key motivation of AGC is to limit the ratio between the parameter update magnitude and the parameter norm ($\|\Delta W^l\| / \|W^l\|$), since large relative updates can cause instability. In the original paper, AGC was implemented with the SGD optimizer, where this effect is achieved by directly clipping gradients. Since SGD is rarely used for large language model pretraining, we adapted AGC’s core idea to AdamW, implementing it as update clipping to achieve equivalent stability.
> > >
> > > ---
> > >
> > > **Q8: Why was LAMB (listed in Table 1) excluded from the experiments?**
> > >
> > > A8: We excluded LAMB from our main experiments for three reasons: (a) LAMB, AGC, and Clippy are all methods that constrain or scale parameter updates, so we chose AGC and Clippy as representative update-clipping baselines; (b) Clippy’s original paper (Table 2) compared LAMB and AGC, showing Clippy performs better; (c) LAMB is an optimizer and, for optimizer-level experiments, we selected the more recent SPAM method for comparison.
> > >
> > > ---
> > >
> > > **Q9: In [2], the authors propose a method also based on gradient values, where they clip a portion of the highest coordinates that consistently degrade the gradient. Could you discuss how AdaGC relates to it or perhaps compare it with their method?**
> > >
> > > A9: Method [2] uses a coordinate-wise approach, clipping a fixed percentage of the largest gradient coordinates per step to reduce directional sharpness and accelerate convergence, mainly for SGD. AdaGC, in contrast, employs an EMA-based adaptive threshold per tensor, dynamically responding to historical gradient norm changes. This makes AdaGC robust to rare, catastrophic anomalies, and specifically effective for eliminating loss spikes in large-scale training. AdaGC is optimizer-agnostic and spatially adaptive, while [2] is coordinate-wise and less dynamic. [2] focuses on sharpness reduction in smaller models, whereas AdaGC targets spike suppression in large-scale pretraining.

---

> > > > ### Author Response · Authors · 2025-11-21
> > > >
> > > > **Q10: As mentioned in W1, could you clarify why no spikes are observed with Muon and Lion in Section 5.4 (Optimizer Compatibility)? Do spikes appear when no clipping is applied?**
> > > >
> > > > A10: We acknowledge that the absence of loss spikes in our Muon and Lion experiments may be due to specific dataset, scale, or initialization choices. In the literature, both optimizers are known to exhibit spikes under certain conditions, e.g., the Kimi K2 Technical Report discusses instability for Muon and the need for MuonClip, while Adam-mini (2024) reports loss spikes on GPT-2 125M for Lion even with GlobalGC applied. Our experiments did not observe spikes, but AdaGC still yielded consistent performance improvements.
> > > >
> > > > AdaGC is an optimizer-agnostic gradient clipping method. In this study, our secondary objective is to evaluate the effectiveness of extending AdaGC to the Muon and Lion optimizers, with the potential to replace GlobalGC as the default gradient clipping strategy in the future. To this end, we conducted two sets of extended experiments: (1) on language models, we applied the Muon optimizer to Llama-2 1.3B; (2) on multimodal models, we applied the Lion optimizer to CLIP ViT-Base.
> > > >
> > > > ---
> > > >
> > > > **S1: Figure 1 Labels**
> > > > > You are correct that all plots in Figure 1 correspond to AdamW + GlobalGC training. We have revised the figure caption and axis labels for clarity, and explicitly emphasize that loss spikes persist even with global norm clipping, highlighting the need for more effective adaptive, tensor-wise approaches like AdaGC.
> > > >
> > > > **S2: Visualization**
> > > > > Thank you for the suggestion. We have tried updating the figures using log-scale axes to make spike events more easily identifiable and the comparisons clearer, but the results are similar to the original plots. The reason is that, in order to show the entire training loss dynamics, the maximum and minimum values of the Y-axis are constrained, and the differences in loss between methods become very small in the later stages of convergence. Since it is important to retain the loss spikes, the curves cannot be smoothed, and even with a log scale, the differences are hardly visible. Our solution is to provide zoomed-in plots of the final convergence interval to better highlight these details.
> > > >
> > > > **S3: Lines 036–059**
> > > > > We agree that the sentence in lines 036–059 is overly long. We will restructure this section with shorter sentences and bullet points for clarity and readability.
> > > >
> > > > **S4: Table 1 - Value-based vs Norm-based**
> > > > > We have clarified the distinction between value-based and norm-based clipping more explicitly in the Related Work section and Table 1, as suggested.
> > > >
> > > > ---
> > > >
> > > > **We hope these clarifications address your concerns. Thank you again for your valuable feedback. If our rebuttal has addressed your concerns satisfactorily, we would appreciate it if you could reflect this in your final score.**

---

> > > > > ### Author Response · Authors · 2025-11-28
> > > > >
> > > > > Thank you for your thoughtful and constructive feedback on our work. We have carefully addressed your comments and revised the work and manuscript accordingly. As the discussion period nears its end, we would greatly appreciate any additional questions or points of clarification. If our responses have satisfactorily addressed your concerns, we kindly ask you to consider reflecting this in your score.
> > > > >
> > > > > Thanks again for your time and expertise.

---

### Comment · Area_Chair_TY6u · 2025-11-26
**Request to respond to the authors' rebuttal**

Dear Reviewers,

Thank you very much for the time and effort you have already dedicated to reviewing this submission.

The authors have provided detailed responses to your comments and the specific questions you raised in your initial reviews. As there has been no follow-up since the rebuttal was posted, I would kindly ask you to read the authors’ rebuttal and engage in the discussion.

Thank you again for your contribution to the review process.

Best,
AC

---

### Meta-Review · Area_Chair_Bx2W · 2025-12-23

**Summary:**

This paper proposes AdaGC, a per-tensor gradient clipping rule that adaptively sets each tensor’s clipping threshold using running statistics, aiming to prevent loss spikes during large-scale pretraining. Reviewers generally agree the problem is practically important and the method is simple to implement, but the decision hinges on whether the work goes beyond a useful optimizer tweak. The core weakness is that the method is essentially a normalization/clipping scheme at the optimizer level, and the paper does not provide a convincing explanation, either empirical or theoretical, of what causes spikes, why AdaGC addresses the cause, or how/when it provably reduces spikes beyond improved stability under selected setups.  As a result, the contribution seems an incremental engineering trick rather than a principled, well-grounded advance, and I thus cannot recommend acceptance.

**Reviewer Concerns:**

The authors added the missing baseline, extended scale evidence (including a longer run), and clarified comparisons and implementation details. These changes improve completeness, but they do not resolve the more fundamental concern that the paper lacks a clear causal story for spikes and lacks strong evidence that AdaGC targets that mechanism rather than serving as an additional normalization heuristic.

**Reviewer Scores:**

All reviewers may remain borderline after rebuttal. Some may increase score from 4 to 6. Overall, the paper remain borderline.

---

### Decision · Program_Chairs · 2026-01-26

Reject